**Implementation of WRF-Hydro at two drainage basins in the region of Attica, Greece for operational flood forecasting**

Elissavet Galanaki[1], Konstantinos Lagouvardos[1], Vassiliki Kotroni[1], Theodore Giannaros[1], Christos Giannaros[2, 1]

[1] National Observatory of Athens, Ioannou Metaxa & Vas. Pavlou, 15236 Penteli, Greece

[2] Frederick Research Center, 7 Filokyprou Street, 1036 Pallouriotisa, Nicosia, Cyprus

Correspondence to: galanaki@meteo.noa.gr

**Abstract.** An integrated modeling approach for forecasting flood events is presented in the current study. An advanced flood forecasting model, which is based on the coupling of hydrological and atmospheric components, was used for a

twofold objective: first to investigate the potential of a coupled hydrometeorological model to be used for flood forecasting at two medium-size drainage basins in the area of Attica (Greece) and second to investigate the influence of the use of the coupled hydrometeorological model on the precipitation forecast skill. For this reason, we used precipitation and hydrometric in-situ data for six flood events at two selected drainage regions of Attica. The simulations were carried out with the Weather Research and Forecasting (WRF) model (WRF-only) and the WRF-

Hydro system in a fully coupled mode, under which surface, subsurface and channel hydrological processes were parameterized at a fine resolution grid of 95 m approximately. Results showed that the coupled WRF-Hydro system was capable to produce the observed discharge during the flood episodes, after the adequate calibration method applied at the studied basins. This outcome provides confidence that the model configuration under the two-way atmospheric-hydrological coupling is robust and, thus, can be used for operational flood forecasting purposes in the area of Attica.

Besides, the WRF-Hydro model ~~has the~~ showed a tendency to slightly improve the simulated precipitation in comparison to the precipitation produced by the atmospheric only version of the model (WRF), demonstrating the capability of the coupled WRF-Hydro model to enhance the precipitation forecast skill for operational flood predictions.

### 1. Introduction

Floods are among the most common natural disasters which related to deaths, destruction and economic losses. Worldwide, 500.000 deaths due to floods have been reported from 1980 to 2009, whit more than 2.8 billion people being affected (Doocy et al., 2013). Petrucci et al. (2018) who developed a flood mortality database in five study areas in the Mediterranean (including Greece) for the period 1980-2015 have found an increasing trend of flood fatalities during the studied period. In Greece and especially in its capital, Athens, flooding events were responsible for 182

deaths from 1880 to 2010 (Diakakis et al., 2013). Papagiannaki et al. (2013) who developed a data base of high-impact weather events over Greece for the period 2001-2011, which is continuously updated since then, showed that flash floods constitute the most common weather-related phenomenon with damages in Greece. Recently, a devastating flash flood, which affected Mandra (in the western Attika region) on 15 November 2017, resulted in 24 deaths and great economic losses, highlighting the consequences of urbanization, uncontrolled constructions and changes in land-use.

Hydrological regimes are affected from climate change. In particular, an increase in the intensity and the frequency of floods, due to human-induced climate modifications, has been reported in the literature (Falter et al., 2015; Wu et al., 2014; Romang et al., 2011; Mily et al., 2002; White et al., 2001).

Given the rapid urbanization, the land-use changes and the human-induced climate change, the risk from future floods is significant and, thus, reliable and accurate flood forecast systems applied over vulnerable areas consists an urgent need.

Flood forecasting strengthens the preparedness phases of disaster management, providing a reduction of the impacts of severe rain events. A reliable and effective flood forecasting system should provide an accurate reproduction of both rainfall and hydrological response inside the targeted drainage areas. In this direction, simulating the land-atmosphere

interactions through coupling of hydrological and atmospheric models, in order to consider the channel and terrain routing of the surface and subsurface water flows, plays an important role (Larsen et al., 2016; Hauck et al., 2011). The terrestrial hydrological processes affect soil moisture, a variable that is crucial for the computation of the sensible and latent heat fluxes, which in turn affect the atmospheric response (Seneviratne et al., 2010; Maxwell et al., 2007). Several studies have shown that an improvement, although not always significant, on the forecasting of the spatiotemporal distribution of extreme synoptic and convective precipitation is provided through the use of coupled hydrometeorological models (e.g., Senatore et al., 2015; Shrestha et al., 2014; Maxwell et al., 2007). Although the mechanisms of the land-atmosphere and hydrology coupling that influence the forecast skill of precipitation is still under investigation, it is well accepted that coupled hydrometeorological models show a significant potential for effective flood forecasting (e.g., Givati et al., 2016). WRF-Hydro, an enhanced version of the Weather Research and Forecasting (WRF) model, is one of the various modeling systems that provides a two-way coupling between the hydrological and land-atmosphere processes. More specifically, it parameterizes overland and river flow routing, subsurface routing in the 2-m soil column, while it also includes a groundwater bucket model, providing, thus, a feedback between terrestrial hydrology and land-atmosphere interactions in the WRF system. The WRF-Hydro model has been used in numerous flood-related research applications (Senatore et al., 2020; Papaioannou et al., 2019; Varlas et al., 2019; Avolio et al., 2019; Lin et al., 2018; Silver et al., 2017; Xiang et al. 2017; Arnault et al., 2016; Givati et al., 2016; Wagner et al., 2016; Senatore et al., 2015; Yucel et al., 2015) and for operational flood forecasting in the United States (Krajewski et al., 2017; NOAA, 2016) and Israel (Givati and Sapir, 2014).

Considering the increased risk and impacts of flooding (Papagiannaki et al., 2013; Diakakis et al., 2012), a reliable flood forecasting system serving operational needs constitutes an urgent need in Attica, where the 36% of the total population lives, while changes in land use and high rates of urbanization are major problems (from 1961 to 2001, the city of Athens increased in size by 82%). This need motivated the present study, which has a twofold objective. Firstly, the investigation of the ability of a two-way coupled hydrometeorological model (WRF-Hydro) to be used for flood forecasting purposes at two drainage basins in the area of Attica after adequate calibration and validation. Secondly, the examination of the influence of the use of the (WRF-Hydro model on the precipitation forecasts as compared to the simulations performed with WRF model.

The next sections of this paper are structured as follows: Section 2 provides a detailed presentation of the methodology followed for the model calibration and the datasets used, Section 3 discusses the results and finally Section 4 hosts the conclusions and the future prospects of this study.

## 2. Methods

### 2.1 Study Area and Data

The study area is the greater area of Attica basin where the capital and largest city of Greece, Athens, and the largest port of the country, Piraeus, are located. Attica basin has an area of 450 km$^2$ and is characterized by a complex geomorphology (Fig. 1a). It is a triangular peninsula with the Cithaeron mountain range to the north acting as a physical division from Boeotia. The population of Attica is ~3.800.000 people (about 36% of the national total) and includes a great part of the national financial and commercial activities.

Papagiannaki et al. (2013) and Diakakis et al. (2012) provided evidences that Attica is the most affected area in Greece concerning weather related hazards and particularly by flash floods. Flash flood events in Attica have been studied from the meteorological point of view (Lagouvardos et al., 1996; among others), the climatological aspect (Galanaki et al., 2018; Galanaki et al., 2016), flood risk (Lasda et al., 2010; Kandilioti and Makropoulos, 2012) and vulnerability (Papagiannaki et al., 2015; 2017). Papagiannaki et al. (2015), in particular, found that impacts of floods increase

significantly when 24-h accumulated rainfall exceeds 60 mm.

In the current study, we focus on two drainage areas of the flood-prone Attica region. The first one is the Sarantapotamos Basin (Figs. 1a, 1c) that drains an area of 310 $km^2$ and is responsible for flooding events in the urbanized broader area of Thriassion plain, located in west Attica, Greece. Among the most important natural flood causes in the area are the geomorphological characteristics of the drainage network, the intense rainfall and the

increasing urbanization which is deprived of integrated flood defense measures. In particular when heavy rainfall occurs, the relatively mild slopes result in a decrease of the surface runoff velocity, accumulating a large volume of water in short times (Zigoura et al., 2014).

The second study area is the Rafina Basin, in Eastern Attica (Figs. 1a, 1d). It drains an area of almost 120 $km^2$ (Karympalis et al., 2005) bounded to the north and northeast by the Penteliko Mountain and to the west and southwest

by the Ymittos Mountain. The area of Rafina was characterized by a rapid residential development over the last decades. In addition, the recent fires, which have burned a significant part of the catchment area, combined with the deflection of Halandri's stream (during the construction of the "Attiki Odos" highway), intensified and increase the frequency of floods in the region (Papathanasiou et al., 2015). It is important to notice that both the studied basins are medium-size catchments (<310 $km^2$) that makes the hydrological simulation challenging.

The data for stage and discharge for Sarantapotamos basin were provided at 15 min intervals from the hydrometric stations of Deucalion project (Fig.1a; http://deucalionproject.itia.ntua.gr), while the data for Rafina basin were derived from the Hydrological Observatory of Athens of National Technical University of Athens. For the meteorological evaluation of the conducted simulations, 10 min precipitation measurements were obtained from the network of surface meteorological stations operated by the National Observatory of Athens (NOA, Lagouvardos et al., 2017). The nearest

meteorological stations to the hydrometric stations used, which are located in Vilia and N. Makri in Sarantapotamos and Rafina, respectively (Fig. 1a).

Six flood events have been considered for the analysis. Table 1 includes the simulation periods of each event, which were selected after spin-up sensitivity experiments (section 2.2.1), and their observed total rainfall and maximum discharge as they have been recorded at the meteorological and hydrometric stations. All examined episodes were

associated with synoptic atmospheric circulation, driven by low-pressure systems, which, in most cases, were combined with 500-hPa troughs and cut-off lows. In particular, surface low-pressure systems, found west of Greece, affected the country in combination with upper-level cut-off lows on 6 February 2012 (event #3) and 29 December 2012 (event #4). In the course of events #1 and #6, the atmospheric circulation was characterized by troughs in the middle troposphere over Greece, associated with surface cyclones located west of North Italy (event #6) and in the Ionian Sea (event #1).

The systems induced considerable precipitation in Greece during the above episodes-resulting to noticeable impacts over the examined basins (Giannaros et al., 2020). The higher impacts in Sarantapotamos catchment were reported in Vilia at the night between 21 and 22 February 2013 (event #5), when 24-h precipitation and maximum discharge reached up to 77 mm and 19.2 $m^3/s$, respectively. During this episode, a very deep surface low crossed the Mediterranean Sea towards Greece. The system was associated with an upper-level trough having a negatively tilted

axis (Giannaros et al., 2020). Between 02 and 05 February 2011 (event #2), exceptional atmospheric conditions affected Greece (Giannaros et al., 2020). Significant impacts were evident in Rafina catchment, where the total 48-h rainfall surpassed 123 mm in N. Makri and the maximum discharge exceeded 24 $m^3/s$ in Rafina. As highlighted above, the events #2 and 5 affected the examined areas more severely and were the most devastating for the whole area of Attica, where floods, deaths, destruction and great economic losses were induced. More details on the hydrometeorological and

socio-economic characteristics of events #2 and #5 can be found in Giannaros et al. (2020).

## 2.2. The Fully Coupled Modeling System

### 2.2.1. Advanced Research WRF

The Advanced Research Weather Research and Forecasting model Version 3.9.1.1 was used in this study (Skamarock et al., 2008) for the land-atmosphere simulations which were carried out using four two-way nested grids (Fig. 1b): d01, d02, d03 d04 with 18 km (325 × 285 grid points), 6 km (685 × 337 grid points), 2 km (538 × 499 grid points) and 667 m (208 × 184 grid points) grid increments, respectively. The coarse domain (d01) encompasses the area of Europe. The higher resolutions domains cover the area of Mediterranean (d02) and Greece (d03), while the finest resolution grid covers the area of Attica. Each domain has 40 unevenly spaced full sigma layers in the vertical direction and the model top was set at 50 hPa. For domains 1, 2 and 3, the 30-arc-sec spatial resolution United States Geological Survey (USGS) GTOPO30 terrestrial data and the 30-arc-sec spatial resolution Moderate Resolution Imaging Spectroradiometer - International Geosphere-Biosphere Project (MODIS-IGBP) global land cover data, have been used. Despite, the high spatial resolution of the MODIS-IGBP dataset, it only includes one category for the urban areas. The latter datasets are considered to be inadequate for ultrahigh-resolution (< 1 km) modeling (Giannaros et al., 2018; Nunalee et al., 2015) , which is necessary for hydrometeorological forecasting (e.g., Verri et al., 2017).Thus, the high resolution Shuttle Radar Topography Mission (SRTM) 90 m × 90 m topography data and the 3-arc-sec resolution Corine Land Cover (CLC) dataset were used for a better land use and topography representation in the innermost d04 domain.

The WRF parametrization schemes used for the simulations are given in Table 2. The selection of the physics schemes was based on sensitivity tests conducted for the exploration of the best-performing schemes in terms of precipitation forecasting in the framework of setting up the model for operational forecasting in Greece. For the cloud microphysics processes, the WRF Single-Moment 6-Class Microphysics scheme (WSM6; Hong and Lim, 2006) was used, which has been also implemented in other studies over Greece (e.g. Emmanouil et al., 2021; Politi et al., 2018; Giannaros et al., 2016; Pytharoulis et al., 2016). The shortwave and longwave radiation fluxes were parameterized with the Dudhia (Dudhia, 1989) and the Rapid Radiative Transfer Model (RRTM; Mlawer et al., 1997) schemes. For the surface layer parameterization the Eta geophysical fluid dynamics laboratory (GFDL) scheme (Schwarzkopf and Fels, 1991) was adopted. The Noah land surface model scheme (Chen and Dudhia, 2001) and Mellor–Yamada–Janjic (MYJ) parameterization (Janjic, 2002) were chosen as land surface and planetary boundary layer schemes, respectively. Noah-MP introduces multiple options and tunable parameters to simulate the land surface processes. However, the default values of these options and parameters are not suitable for every study area (e.g. Giannaros et al., 2019). In contrast, the Noah LSM has been tested and applied successfully in several studies focusing in Greece (e.g. Varlas et al., 2019; Papaioannou et al., 2019; Giannaros et al., 2020). In addition, MYJ parameterization scheme has been successfully implemented in other studies over Greece (e.g. Emmanouil et al., 2021; Politi et al., 2018). Cumulus parameterization, namely the Kain-Fritch scheme (Kain et al., 1992), was activated only for d01 and d02.

The simulations were initialized and forced at its lateral boundaries by meteorological data derived from ERA5 reanalysis data (Hersbach and Dee, 2016) provided by the European Center for Medium-Range Weather Forecasts (ECMWF). The reanalysis data have a spatial resolution of 0.25° × 0.25°, having 37 pressure levels in the vertical direction and are provided at 6 h intervals. It should be noted that the use of ERA5 reanalysis data was preferred instead of the operational GFS data, as the on-line availability of the GFS forecasts is limited for historical periods. GFS initialization data could be ordered for the investigated events but at a high spatial resolution of 0.5° × 0.5°, which was not considered adequate for forcing the WRF simulations having a coarse domain (do1) resolution of 18 km.

Using the aforementioned setup, a series of sensitivity tests were performed in order to explore the best spin-up time for each event. Precisely, four numerical simulations were conducted for each event, starting at 24h, 18h, 12h and 6h before the initiation of the rainfall. The choice of the best spin-up time for each simulation was made by comparing the

temporal evolution of precipitation produced by the WRF model with the observed precipitation at the rain gauge
station at Vilia for the basin of Sarantapotamos and at the rain gauge station at N. Makri for the Rafina basin. An
example of the temporal evolution of the rainfall in Vilia for event #5 is given in Fig. 2.

### 2.2.2. WRF - Hydro

The WRF-Hydro modeling system, version 3.0, was used for this study under a fully coupled mode. WRF-Hydro is a
distributed hydrometeorological modeling system which is two-way coupled with WRF, providing multiple physics
options for surface overland flow, saturated subsurface flow, channel routing, and base-flow processes (Gochis et al.,
2015). The main advantage of WRF-Hydro is the ability to simulate the specialized components of water cycle such as
soil moisture and ground water, considering the routing processes of the infiltration capacity excess and the saturated
subsurface water.
In the present study, the WRF-Hydro was configured for the d04 domain, in a coupled manner with physics options of
surface flow, sub-surface flow and channel routing activated. The catchments' routing grids were computed based on
SRTM 90 m topography data using the WRF-Hydro GIS pre-processing toolkit. In order to exploit this high-resolution
input dataset, avoiding interpolation to a coarser grid (Verri et al., 2017; Gochis and Chen, 2003), a ~95 m spatial
resolution WRF-Hydro domain was configured over the WRF innermost domain. Thus, the ratio between the high-
resolution terrain routing grid and the WRF land surface model (aggregation factor; AGGFACTRT) was set to 7. The
soil water infiltration and redistribution was computed in 4 layers (0–10, 10–40, 40–100, and 100–200 cm) in the fine
resolution grid and then was aggregated in the coarser grid of d04.
Subsurface lateral flow of soil was calculated by applying the methodology proposed by Wigmosta et al. (1994) and
Wigmosta and Lettenmaier (1999) prior to the routing of overland flow, allowing the exfiltration from fully saturated
grid cells to be added to the surface flow of the coarser grid. The effects of topography and the saturation depth of soil
were included in the calculation of subsurface flow. Thus, when the depth of ponded water on a grid cell exceeded a
threshold, the overland flow was solved with a diffusive wave formulation adapted from Julien et al. (1995) and Ogden
(1997).

**2.3 Calibration method**

The aim of the WRF-hydro calibration is to improve the spatial resolution of parameters that control the total water
volume and the shape of the predicted hydrograph. Generally, the calibration processes for WRF-Hydro can be divided
into three categories: the manual step-wise (e.g. Li et al., 2017), the automate calibration process and mixed calibration
approaches combining manual and automate calibration (e.g. Verri et al., 2017). The step-wise approach of calibration is
widely applied in order to minimize the high number of model runs and are required for the automate calibration
approach.
WRF-Hydro has numerous tabulated parameters that influence the simulated hydrological processed and the output
discharge. Yucel et al. (2015) showed that four parameters are the most critical for the simulated hydrograph. Thus, in
this study, calibration procedure was based on the stepwise method suggested by Yucel et al. (2015), and implemented
by other authors also (e.g. Wang et al, 2020). The stepwise calibration was performed in two basic steps: firstly, we
defined the parameters that influence the total water volume and then we calibrated the parameters controlling the shape
of the hydrograph. The parameters that control the total water volume, are the runoff infiltration factor (REFKDT) and
the surface retention depth (RETDEPRTFAC). The REFKDT parameter controls the amount of water that flows into the
channel network, while the RETDEPRTFAC influences the surface slope and thus the accumulation of the water.

The parameters that control the shape of the hydrograph, are related to the surface (OVROUGHRT) and channel
roughness (Manning's roughness, MannN). Thus, the parameters were calibrated in the following order: REFKDT,
RETDEPRTFAC, OVROUGHRTAC and MannN. The parameters are abbreviated following the nomenclature of WRF-
Hydro namelist. The calibrated values for each parameter are shown in Table 3, along with the default values. MannN
parameter is defined for each stream order in the drainage area. The ArcGIS pre-processing tool, used for the

reproduction of the hydrological features of the studied catchments resulted in four Strahler stream orders (Strahler,
1957) in both Sarantapotamos and Rafina. Thus, MannN values in Table 3 are shown for the first four stream orders. In
the stepwise calibration method, sensitivity tests were performed for each parameter and when a parameter was
calibrated its optimum value was kept constant when the sensitivity tests for the next parameter were performed.
Further details on the calibration of the aforementioned parameters for each basin (Sarantapotamos & Rafina) are given

in the following section. The calibration of the WRF-Hydro model was performed using the WRF atmospheric forcing,
including the precipitation fields, following the same approach of forcing the model with WRF data from previous
studies (e.g. Li et al. 2020; Liu et al., 2020; Li et al. 2017).

### 3. Results and Discussion

**3.1. Sarantapotamos basin**

**3.1.1. Calibration of Sarantapotamos basin**

Due to limited availability of streamflow data, the calibration process was performed only for event #5 at the basin of
Sarantapotamos, while the rest of the events were used to evaluate the performance of the calibration process. Fig. 3
shows the evolution of the discharge (observed and simulated) for each calibrated parameter. The choice of the optimum

value for each parameter was based on the selected objective criteria, namely the Nash-Sutcliffe efficiency and the
correlation coefficient (R), between simulated and observed discharges.

Fig. 3a shows the results for the first parameter of the step-wise calibration method (REFKDT). As possible values for
the REFKDT parameter range from 0.5 to 5, we firstly performed several simulations for possible REFKDT's values of
1, 2, 3, 4 and 5 (not shown) in order to find the appropriate range of the scaling factor. Thus, the appropriate range of

REFKDT was found to be from 0.5 to 1.5 and then additional simulations were performed within this range with
increment of 0.1. Fig. 3a shows that the discharge decreases as the REFKDT's values increase. For the selection of the
optimum value of each parameter we implemented two basics steps. Firstly, a visual comparison of the simulated and
observed discharge was performed. Secondly, we applied statistical analysis tests. More precisely, the statistical analysis
included the computation of correlation coefficient and the Nash-Sutcliffe coefficient between the observed and

simulated discharge calculated per 15 min for each possible value of REFKDT (Fig. 4). Thus, the value which has the
best correlation for the Nash-Sutcliffe coefficient was chosen as the optimum value, after the visual comparison of the
simulated and observed discharge. Namely, the value of 0.5 for REFKDT parameter was selected. Table 4 shows the
correlation and the Nash–Sutcliffe coefficient for the optimum value for each parameter

It is noted that there is a lag at the time of maximum discharge between the observations and the model results. This

discrepancy is attributed to the time lag between the simulated and observed temporal evolution of precipitation at Vilia
station (Fig. 2). After the implementation of cross correlation analysis, it was found that the maximum correlation
between the simulated and the observed temporal evolution of precipitation is achieved with a delay of 5 hours. It must
be noted that the results of the statistical analysis presented in Table 4 are computed after the displacement of the
temporal evolution of the simulated discharge. This displacement of 5-h was necessary in order to derive the optimum

value of each parameter. For instance, if we do not take into account the 5-h gap, the correlation and the Nash-Sutcliffe

coefficients are not in the acceptable limits, thus the choice of the optimum value for each parameter cannot be determined.

Fig. 3b shows the temporal evolution of discharge for the possible RETDEPRTFAC values. The possible values of RETDEPRTFAC range from 0 to 10, while an increment of 1 was used for the simulations. The RETDEPRTFAC is related to the retention depth of water from the surface. Thus, if the RETDEPRTFAC value is 0, there is no accumulation of water in the area. Fig. 3b shows that the simulated discharge is decreasing with increasing values of RETDEPRTFAC. The value of 10 for RETDEPRTFAC parameter was selected based on visual comparison of the model and observed discharge and on the statistical analysis, following the aforementioned procedure for the selection of REFKDT (not shown). It should be noted that the optimal parameters for REFKDT and RETDEPRTFAC hit the lower and calibration limit, respectively. Relaxing their constraints may result to better calibrations results.

Figs 3c and 3d show the temporal evolution of discharge for the parameters which control the hydrograph shape (OVROUGHRTAC and Manning's roughness). The OVROUGHRTAC parameter is related to the surface roughness of the channel and was calibrated for values between 0.1 and 1.0 with 0.1 increments (Fig. 3c). Finally, a scaling factor value of 0.4 for OVROUGHRTAC parameter was selected.

As Manning coefficient values are based on textbook values for each stream order, Yucel et al. (2015) suggested multiplying the default MannN coefficient parameter with a scaling factor. Fig. 3d shows the temporal evolution of discharge for the possible values of MannN scaling factors ranging from 0.6 to 2.1 with increments of 0.1. Finally, the value of 1.1 was selected as optimum for MannN parameter.

**3.1.2. Validation of the calibration of Sarantapotamos basin**

After the calibration of WRF-Hydro over Sarantapotamos basin based on the event #5, the four parameters defined above were validated for the events #4 and #6 of Sarantapotamos basin. Figures 5b and 5d show the comparison of the temporal distribution of the observed and simulated discharges for the events #4 and #6, respectively. For the event #4, the simulated temporal distribution of the discharge shows similarity to the observed one (Fig. 5b), as the time that the maximum discharge occurred almost coincides while the two temporal distributions don't show similar maximum values of discharge (the observed discharge is 12.8 $m^3$/s and the simulated is 5.7 $m^3$/s).

The correlation coefficient of the two temporal distributions is 0.83. For the event #6, the simulated and observed temporal distribution of the discharges show similarity in the occurrence time of the maximum discharge values but the simulated discharge underestimates the observed one throughout the duration of the event (Fig. 5d), as the maximum value of the simulated discharge is 10.6 $m^3$/s while the observed one is 7 $m^3$/s. This is due to the underestimation of the simulated rainfall at the station of Vilia compared to the observed one (Fig. 5c). The correlation coefficient between the simulated and observed discharges is 0.84.

**3.2. Rafina basin**

**3.2.1. Calibration of Rafina basin**

The stepwise calibration method suggested above, was implemented for the calibration of Rafina basin using event #2. Figure 6 shows the temporal distribution of the precipitation as observed at the station of N. Makri and simulated using WRF atmospheric only simulations and WRF-Hydro coupled simulations, while Fig.7 shows the temporal evolution of the observed and simulated discharges for the possible values of each calibrated parameter. The observed and simulated precipitation (provided by WRF-Hydro) are highly correlated (correlation coefficient: 0.83) while quantitatively they also compare very well (Fig. 6). The choice of the optimum values for each parameter was based on the visual

comparison of the simulated and observed discharge (Fig. 7) and statistical analysis (Table 5), as it was explained for Sarantapotamos basin.

In consistency to the calibration of Sarantapotamos, we firstly performed several simulations for possible REFKDT's values between 1 and 5 and we also found that the appropriate range of the scaling factor from 0.5 to 1.5. Thus, the additional simulations were performed within this range with increment of 0.1 and the value of 0.5 was selected as optimum value for REFKDT parameter. As in the case of Sarantapotamos, the optimum value for REFKDT reaches the lower calibration limit indicating that changing the calibration limit may let to better result. The simulations for RETDEPRTFAC were performed within the range from 0 to 10, with increment of 1. As in the case of Sarantapotamos, the simulated discharge is decreasing with increasing values of RETDEPRTFAC (Fig. 7b). After the comparison of the aforementioned statistical criteria, the selected optimum value for the RETDEPRTFAC parameter was 6.

Regarding the parameters controlling the shape of the hydrograph, 10 (from 0.1 to 1.0 with increment of 0.1) and 1.6 (from 0.6 to 2.1 with increment of 0.1) simulations performed for the parameters related to the surface and channel roughness, respectively. After the computation of correlation coefficient and Nash-Sutcliffe parameter for each simulation, the optimum values of 0.3 and 1.2 for OVROUGHRTAC and MannN parameters were selected. At the end of the calibration procedure, the two temporal distributions (observed and discharge) have correlation coefficient 0.62, while the Nash–Sutcliffe is close to 0.5 (Table 5).

### 3.2.2. Validation of the calibration of Rafina Basin

The validation of the calibration process of Rafina basin was held by comparing the temporal distributions of simulated and observed discharges of the events #1, #3 and #4 (Figs 8b, 8d and 8e), using the optimum values of the calibration's parameters. The correlation coefficients between the simulated and observed discharges are 0.77, 0.86 and 0.62, respectively. Therefore, it is obvious that WRF-Hydro is capable to forecast the discharge after the calibration process. The simulated discharge is dependent on the simulated precipitation, thus a possible underestimation of the simulated discharge is influenced by a possible underestimation of the precipitation. For instance, at event #1, the maximum simulated discharge is 5.0 $m^3$/s while the observed one is 8.0 $m^3$/s (Fig. 8b). This is attributed to the underestimation of the total precipitation, as the total simulated precipitation is 27.6 mm while the observed is 37.0 mm. Besides, the lag between the observed and simulated discharge is attributed to the lag of the observed and simulated precipitation (Fig. 8a).

### 3.3 Precipitation

In this section the influence of the use of the coupled model (WRF-Hydro) on ~~the improvement of~~ the precipitation forecast skill as compared to the atmosphere-only simulations performed with WRF model will be investigated. Namely, WRF-Hydro contributes to a better simulation of the soil moisture content, due to the computation of the lateral redistribution and re-infiltration of the water (Gochis et al., 2013). The improved simulation of the soil moisture affects the computation of the sensible and latent heat fluxes, which influence humidity and temperature in the lower atmosphere and consequently precipitation (Seneviratne et al., 2010). Therefore, the physical process of the coupling of land-atmosphere is expected to improve the forecast skill of precipitation.

Figures 2, 5a, 5c, 6, 8a, 8c and 8f show the temporal distribution of the precipitation observed and simulated by WRF only and WRF-Hydro for each studied event observed in Sarantapotamos and Rafina basins, for the gauge stations in Vilia and N. Makri respectively. In all cases, the precipitation reproduced by WRF-Hydro has differences compared to WRF (atmospheric only) simulations. The temporal distribution of WRF-Hydro and WRF follow the same pattern as this is reflected in the same calculated correlation coefficients shown in Table 6. WRF-Hydro performs better than the

WRF in terms of quantitative precipitation forecasting and this is reflected to the lower calculated Root Mean Square Errors and the lower Mean Absolute Error (MAE), which have been computed based on the hourly values of precipitation (Table 6). It must be noted that for events #1 and #4 despite the fact that the correlation coefficient is low, due to the lag between simulated and observed discharge (Figs. 8b and 8f), the values of total amount of the simulated and observed precipitation are similar. Also, the low correlation coefficient and the high MAE at event #5 is attributed to the time lag between the simulated and observed temporal evolution of precipitation (Fig. 2).

Fig. 9 shows the difference between the total amount of precipitation observed minus the total amount of precipitation simulated by a) WRF-Hydro and b) WRF-only for each event. Therefore, values close to zero mean that the total amount of precipitation simulated is close to the observed one. For each case, the difference between the total amount of observed and simulated precipitation by WRF-Hydro is smaller pointing out that WRF-Hydro has the tendency to improve the total amount of precipitation, in consistency to the results provided by Givati et al. (2016) and Wang et al. (2020).

Table 7 shows the basin average soil moisture (at the $1^{st}$ level) and latent heat flux simulated by the WRF-Hydro and WRF-only models, at the time before the beginning of the examined storms events. As can be seen the soil moisture differences between the models range from 0.005 to 0.027 $m^3$ $m^{-3}$ and latent heat flux differences span from 0.038 to 16.862 $W/m^2$. These differences simulated by the two models provides an indication that the most accurate replication of the observed precipitation provided by the WRF-Hydro model compared to the WRF-only model is related to the physical process associated with the coupling of land-atmosphere and hydrological routing in the WRF-Hydro model. In particular, WRF-Hydro, affects the soil moisture content, due to the computation of the lateral redistribution and re-infiltration of the water (Gochis et al., 2013), which in turn influences the computation of the sensible and latent heat fluxes. These fluxes are associated with humidity and temperature in the lower atmosphere and consequently precipitation (Seneviratne et al., 2010). However, it should be noted that the effects of soil moisture on precipitation fields are more evident and valid in long-term simulations when the land surface variables reach a steady state (Fersch et al., 2020; Senatore et al., 2015).

## 4. Conclusions

Despite flash flooding is one of the most costly weather-related natural hazards in Greece (Papagiannaki et al., 2013), less effort has been taken place in the field of evaluating tools to predict floods The current paper addresses this issue by presenting an integrated modeling approach for simulating flood episodes in Attica, Greece in medium catchment size basins. The objective of this study was twofold: to investigate the ability of WRF-Hydro to simulate selected cases of flood occurrence in the area of Attica (Greece) and to study the influence of land-atmosphere interactions on the precipitation forecasting. For that purpose, we first calibrated and validated WRF-Hydro at two drainage basins (Sarantapotamos basin and Rafina basin) in the area of Attica. Then, we investigated the relation between WRF-Hydro and WRF-only precipitation forecast skill. For this reason, we used an enhanced version of WRF, the WRF-Hydro model (version 3.0), in a fully coupled mode, which is complemented with the land-atmosphere interaction schemes through the coupling of hydrological and atmospheric models. The configuration of WRF- Hydro was applied in a fine resolution grid (666 m) where the surface and subsurface flow were computed at a grid interval of 95 m.

Three flooding events at Sarantapotamos basin and four flooding events at Rafina basin have been analyzed. Calibration procedure was based in the manual stepwise method proposed by Yucel et al. (2015) defining the parameters REFKDT, RETDEPRTFAC, OVROUGHRTAC and MannN, which influence the total water volume and the shape of the hydrograph. Results showed that the correlation coefficient between the observed and simulated discharges after the calibration was higher than 0.7 for all events. Thus, WRF-Hydro is capable of forecasting observed discharge at the

studied regions, after implementation of a successful calibration process. This outcome is important because WRF-Hydro is implemented under calibration with ground-truth observations for the first time in Greece, contributing this way to the better modeling and understanding of flooding mechanisms in the study areas. Additionally, these calibrated parameters could be used from every scientific team that wants to study past and future flooding events in the area of

Attica, enhancing the research community's understanding of the physical effects of flash flooding.

To investigate the influence of the use of the WRF-Hydro on the precipitation forecast skill, we compare the simulations produced by WRF-Hydro and WRF-only models, configured with the same microphysics schemes for all events. The resulted simulations were verified against observed precipitation in two gauge stations: at Vilia (for the basin of Sarantapotamos) and N. Makri (for the basin of Rafina). Thus, we compared the simulated against observed

precipitation both in terms of temporal distribution and total amount of precipitation. We found that the temporal distribution of WRF-Hydro simulations has the same correlation coefficient but it has lower root mean square errors than the simulation of WRF-only. Although it was shown that the WRF-Hydro tends to slightly improve the total amount of forecasted precipitation, the overall results indicate that the components of terrestrial hydrological models are contributing but not decisive factors in the simulation of precipitation. A preliminary analysis of key water budget

components indicated that the precipitation simulation improvement provided by the WRF-Hydro system may be related to the feedback of the terrestrial hydrology parameterization on the modeled atmosphere. A follow up study could focus on the further investigation of impact of the more detailed representation of the interaction between the land surface and hydrology processes to the surface energy budget under the WRF-Hydro coupling scheme by applying long-term simulations and validated the results against ground-based or satellite observation, considering limitations

arising from internal model variability (Bassett et al., 2020) and domain size (Fersch et al, 2020; Arnault et al., 2018). Also, the incorporation of the SST update into the model will be considered, as previous studies shown a positive feedback to simulations (Avolio et al., 2019; Senatore et al., 2015). Even though a more detailed analysis is required to explore the sensitivity of the simulated precipitation to the coupling between hydrological and land-atmosphere processes, the current study demonstrates that the coupled WRF-Hydro model has the potential to enhance precipitation

forecast skill for operational flood predictions.

For an operational point of view, the application of a coupled WRF-Hydro model to exploit its beneficial impact in simulating precipitation is partially limited due to the additional computational time needed for the execution of the WRF-Hydro model. In particular, in our case, a three day coupled WRF-Hydro forecast considering a prior 12 hours spin up under the investigated configuration requires x1.35 time compares to WRF-only implementation in 140

computing nodes. It should be noted that the extra computational time depends on the WRF-Hydro configuration and the computing resources, in which the model is applied.

It is in our prospects, to further enhance the performance of WRF-Hydro in the study areas and expand the applied modeling approach in other drainage basins throughout Greece, with the aim to build an operational flood forecasting system based on coupled hydrological and atmospheric models. Thus, this work is a preliminary effort in order to

develop a prototype flood forecasting system, based on the state-of-the-art hydrometeorological modeling tool WRF-Hydro, and establish efficient dissemination tools promoting flood-risk awareness. The utmost goal is to provide citizens and stakeholders with information and warnings in order to enhance flood risk awareness and protect lives and properties.

**Data availability**. Data from this research are not publicly available. Interested researchers can contact the corresponding author of this article.

**Author contributions.**

The study was conceptualized by all authors; EG carried out the simulations and wrote the original draft. KL, VK, TG
and CG provided comments for the results, reviewed and edited the manuscript.

**Competing interests.**

The authors declare that they have no conflict of interest.

**Acknowledgements**

The data for stage and discharge for Sarantapotamos basin were provided from the hydrometric stations of Deucalion project (http://deucalionproject.itia.ntua.gr), while the data for Rafina basin were provided from the Hydrological Observatory of Athens of the National Technical University of Athens, kindly provided by Professor E. Baltas. Christos Giannaros received support by the project Cyprus Flood Forecasting System—POST-DOC/0718/0040, which is co-
funded by the Republic of Cyprus and the European Regional Development Fund (through the 'DIDAKTOR' RESTART 2016–2020 Programme for Research, Technological Development and Innovation).

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

**Table 1. Simulation periods of each event and hydrometeorological charecteristics**

| | Basin | Simulation date Start | Simulation date End | Spin-up | Total rainfall | Maximum discharge |
|---|---|---|---|---|---|---|
| Event #1 /E1 | Rafina | 02/01/2011 00:00 UTC | 03/01/2011 18:00 UTC | 6h | 37.6 mm of rain (24 h accumulated) in N. Makri | 8 m$^3$/s in Rafina |
| Event #2 /E2 | Rafina | 02/02/2011 00:00 UTC | 05/02/2011 18:00 UTC | 24h | 123.8 mm of rain (48 h accumulated) in N. Makri | 24.3 m$^3$/s in Rafina |
| Event #3 /E3 | Rafina | 06/02/2012 06:00 UTC | 08/02/2012 18:00 UTC | 6h | 33.6 mm of rain (48 h accumulated) in N. Makri | 9.1 m$^3$/s in Rafina |
| Event #4 /E4R | Rafina | 28/12/2012 06:00 UTC | 31/12/2012 18:00 UTC | 18h | 86.8 mm of rain (72 h accumulated) in N. Makri | 44.3 m$^3$/s in Rafina |
| Event #4/E4S | Sarantapotamos | 28/12/2012 18:00 UTC | 01/01/2013 18:00 UTC | 18h | 104.6 mm of rain (72 h accumulated) in Vilia | 12.8 m$^3$/s in Vilia |
| Event #5 /E5 | Sarantapotamos | 21/02/2013 18:00 UTC | 23/02/2013 18:00 UTC | 6h | 77 mm of rain (24 h accumulated) in Vilia | 19.2 m$^3$/s in Vilia |
| Event #6 /E6 | Sarantapotamos | 02/03/2014 00:00 UTC | 04/03/2014 18:00 UTC | 24h | 85 mm of rain (48 h accumulated) in Vilia | 10.7 m$^3$/s in Vilia |

**Table 2. The WRF Physics schemes used**

|  | Europe (d01) | Mediterranean (d02) | Greece (d03) | Attica Basin (d04) |
|---|---|---|---|---|
| Microphysics | WSM6 | WSM6 | WSM6 | WSM6 |
| Cumulus physics | KF | KF | - | - |
| Shortwave/longwave radiation physics | RRTMG/RRTMG | RRTMG/RRTMG | RRTMG/RRTMG | RRTMG/RRTMG |
| Planetary boundary layer physics | MYJ | MYJ | MYJ | MYJ |
| Surface layer physics | Eta similarity | Eta similarity | Eta similarity | Eta similarity |
| Land surface model | Noah | Noah | Noah | Noah |

**Table 3. The range of calibrated parameters**

| Parameter | Definition | Range of scaling factor | Increment | Default value |
|---|---|---|---|---|
| REFKDT | runoff infiltration | 0.5-1.5 | 0.1 | 3.0 |
| RETDEPRTFAC | surface retention depth | 0-10 | 1 | 1.0 |
| OVROUGHRTAC | surface roughness | 0.1-1 | 0.1 | 1.0 |
| Manning's roughness/ stream order 1 | channel roughness | 0.33-1.16 | 0.1 | 0.55 |
| Manning's roughness/ stream order 2 | channel roughness | 0.21-0.74 | 0.1 | 0.35 |
| Manning's roughness/ stream order 3 | channel roughness | 0.09-0.32 | 0.1 | 0.15 |
| Manning's roughness/ stream order 4 | channel roughness | 0.06-0.21 | 0.1 | 0.10 |

**Table 4. The correlation coefficient and the Nash-Sutcliffe test between the observed hydrograph and the simulations for the optimum values of each parameter for Sarantapotamos basin, after the 5h displacement of the temporal evolution of the simulated discharge.**

| Parameter | Correlation (R) | Nash-Sutcliffe |
|---|---|---|
| REFKDT = 0.5 | 0.86 | 0.67 |
| RETDEPRTFAC = 10 | 0.87 | 0.65 |
| OVROUGHRTAC = 0.4 | 0.89 | 0.69 |
| MannN = 1.1 | 0.85 | 0.67 |


**Table 5. The correlation coefficient and the Nash-Sutcliffe test between the observed hydrograph and the simulations for the optimum values of each parameter for Rafina basin.**

| Parameter | Correlation (R) | Nash-Sutcliffe |
|---|---|---|
| REFKDT = 0.5 | 0.48 | -0.06 |
| RETDEPRTFAC = 6 | 0.38 | -0.6 |
| OVROUGHRTAC = 0.3 | 0.46 | 0.19 |
| MannN = 1.2 | 0.62 | 0.51 |

**Table 6. Comparison of total amount of observed precipitation to WRF-Hydro and WRF only simulated precipitation for each event for the gauge stations in Vilia and N. Makri. RMSE, R and MAE are calculated on hourly values of precipitation.**

| | | Total precipitation | Root Mean Square Error (rmse) | Correlation (R) | Mean Absolute Error (MAE) |
|---|---|---|---|---|---|
| Event #1 /E1 | Rain gauge station in N.Makri | 37.6 | - | - | - |
| | WRF-Hydro | 27.6 | 0.14 | 0.23 | 0.78 |
| | WRF | 51.6 | 0.19 | 0.23 | 1.06 |
| Event #2 /E2 | Rain gauge station in N.Makri | 123.8 | - | - | - |
| | WRF-Hydro | 138.2 | 0.12 | 0.83 | 0.53 |
| | WRF | 92.3 | 0.32 | 0.83 | 1.02 |
| Event #3 /E3 | Rain gauge station in N.Makri | 33.6 | - | - | - |
| | WRF-Hydro | 30 | 0.025 | 0.43 | 0.49 |

| | | | | | |
|---|---|---|---|---|---|
| | WRF | 45.1 | 0.24 | 0.43 | 0.65 |
| Event #4 /E4R (Rafina) | Rain gauge station in N.Makri | 86.8 | - | - | - |
| | WRF-Hydro | 96.6 | 0.12 | 0.2 | 1.64 |
| Event #4 /E4S (Sarantapotamos) | WRF | 85.1 | 1.09 | 0.2 | 2.39 |
| | Rain gauge station in Vilia | 104.6 | - | - | - |
| | WRF-Hydro | 121.3 | 0.3 | 0.57 | 1.83 |
| Event #5 /E5 | WRF | 218.9 | 2.06 | 0.57 | 3.35 |
| | Rain gauge station in Vilia | 77 | - | - | - |
| | WRF-Hydro | 30.2 | 1.06 | 0.13 | 2012 |
| Event #6 /E6 | WRF | 22.1 | 1.2 | 0.13 | 2823 |
| | Rain gauge station in Vilia | 85 | - | - | - |
| | WRF-Hydro | 49 | 0.72 | 0.75 | 1.33 |
| | WRF | 37.7 | 1.03 | 0.75 | 1.43 |


**Table 7. Comparison of the basin average soil moisture (at the 1$^{st}$ level) and latent heat flux simulated by the WRF-Hydro and WRF-only models, at the time before the beginning of the events.**

| | Basin | | Soil moisture ($m^3$ $m^{-3}$) | Latent heat ($W/m^2$) |
|---|---|---|---|---|
| Event #1 /E1 | Rafina | WRF-Hydro | 0.2915 | 21.8365 |
| | | WRF | 0.3034 | 4.9744 |
| Event #2 /E2 | Rafina | WRF-Hydro | 0.2760 | 8.1130 |
| | | WRF | 0.2660 | 8.0754 |
| Event #3 /E3 | Rafina | WRF-Hydro | 0.3427 | 112.7901 |
| | | WRF | 0.3159 | 111.3941 |

| | | | | |
|---|---|---|---|---|
| Event #4 /E4R | Rafina | WRF-Hydro | 0.2126 | -8.8773 |
| | | WRF | 0.2121 | -9.2911 |
| Event #4/E4S | Sarantapotamos | WRF-Hydro | 0.2248 | 43.0125 |
| | | WRF | 0.2316 | 31.2754 |
| Event #5 /E5 | Sarantapotamos | WRF-Hydro | 0.2834 | -3.8582 |
| | | WRF | 0.2823 | -3.9325 |
| Event #6 /E6 | Sarantapotamos | WRF-Hydro | 0.2792 | 8.2810 |
| | | WRF | 0.2666 | 2.9012 |

**Figure 1: (a) Terrain elevation of the studied domain (obtained by MODIS-IGBP global land cover data) along with two channel network and the positions of the meteorological (triangle marker) and hydrometric stations (star marker). (b) Modeling domains. The borders of** *analyzed catchments along with the land cover for* **(c) Sarantapotamos and (d) Rafina basins**

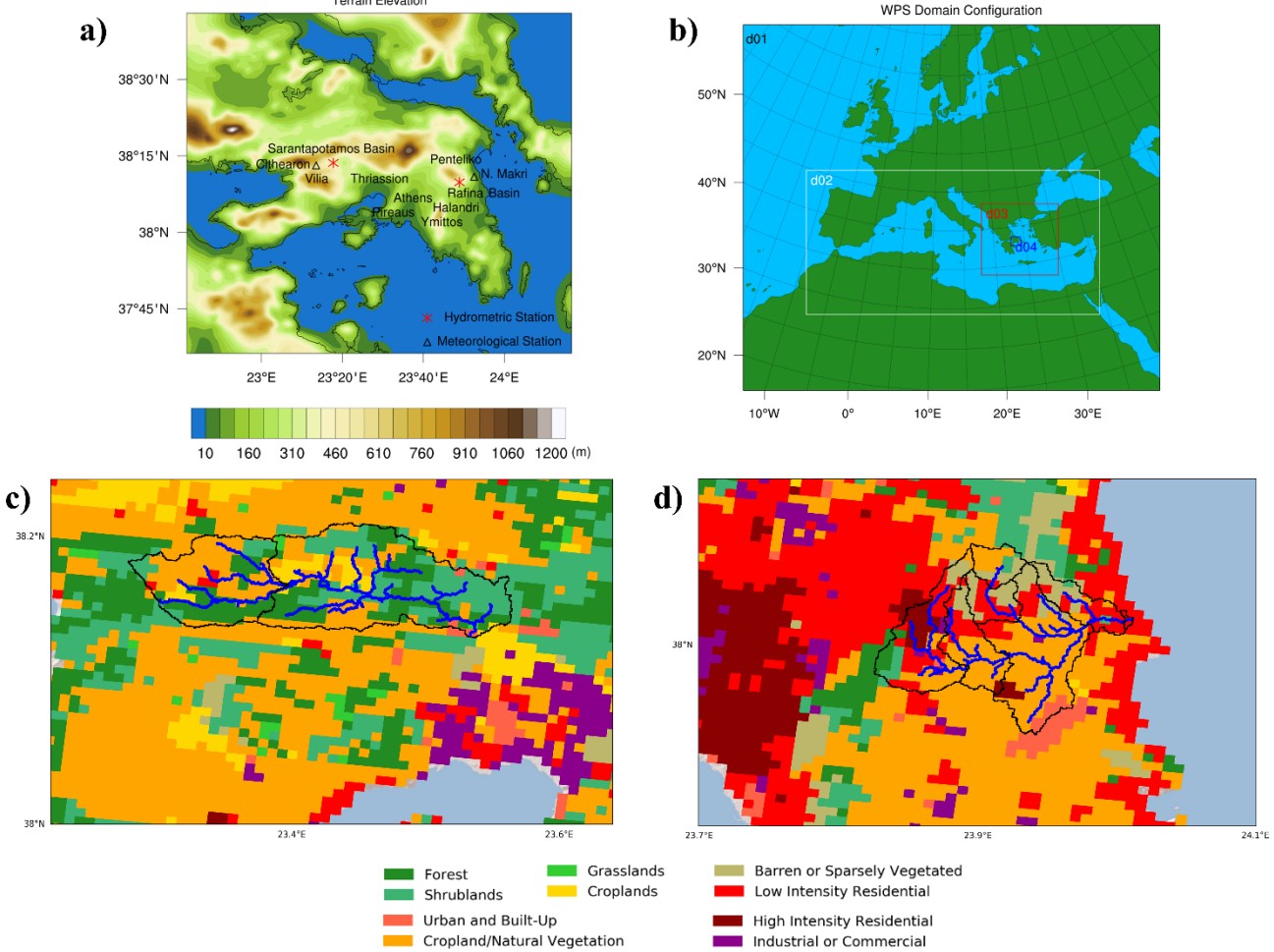

**Figure 2: The temporal evolution of the precipitation in rain gauge station at Vilia for the event #5**


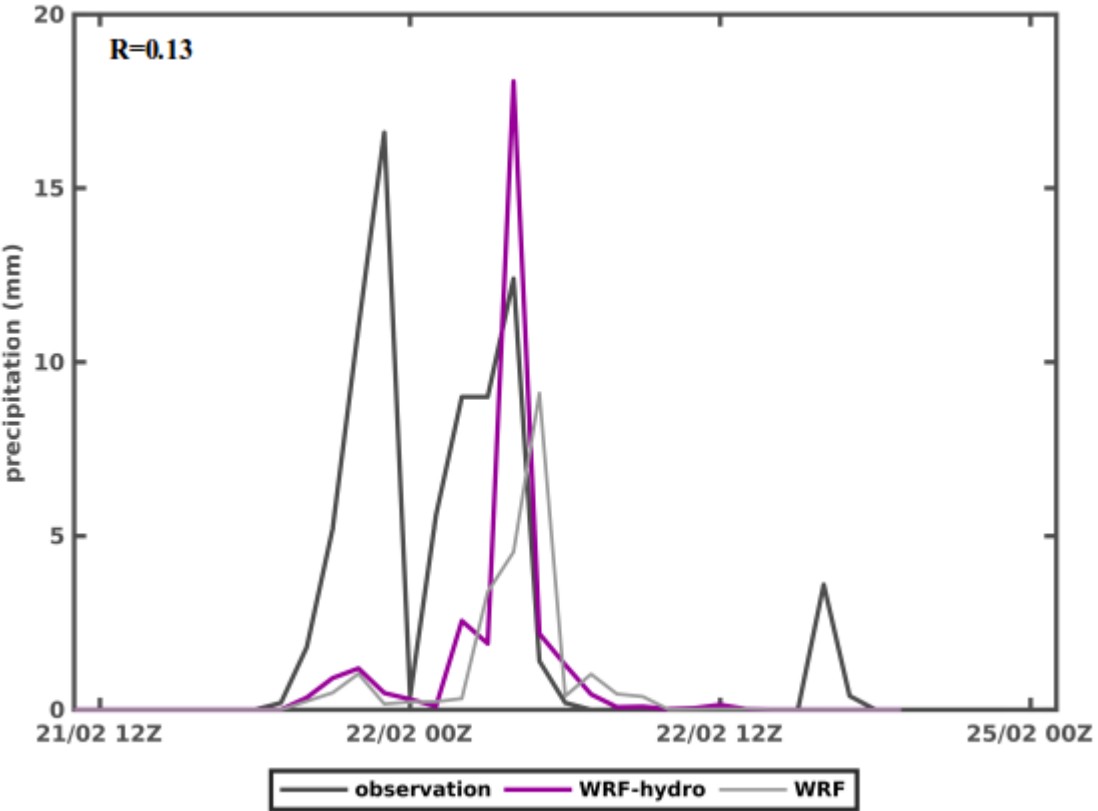

**Figure 3: The evolution of the discharge (observed and simulated) for event #5 for (a) REFKDT, (b) RETDEPRTFAC, (c) OVROUGHRTAC and (d) MannN parameter.**

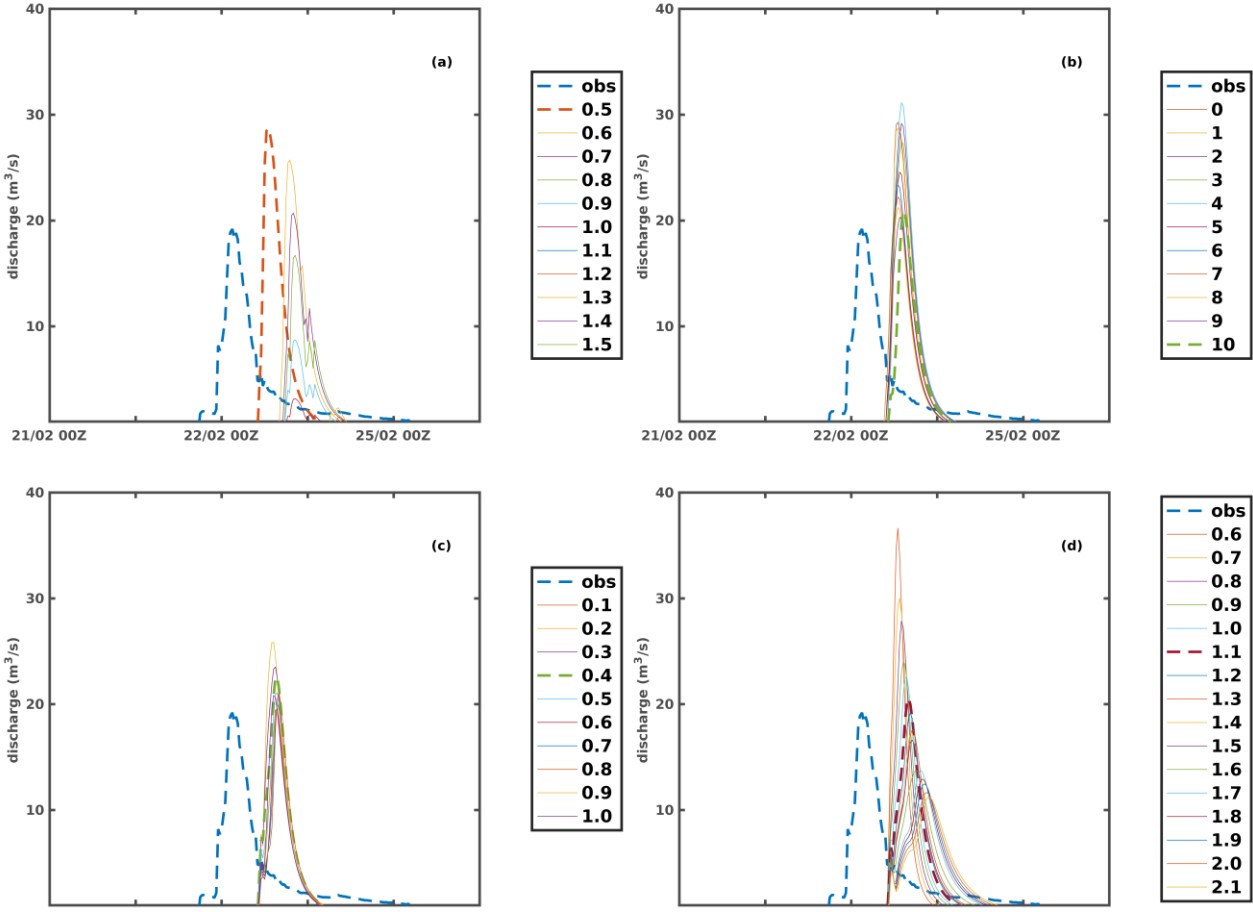


**Figure 4: (a) The correlation and (b) the Nash-Sutcliffe coefficient between the observed and simulated discharge for each possible value of REFKDT for Sarantapotamos basin, after the 5h displacement of the temporal evolution of the simulated discharge.**

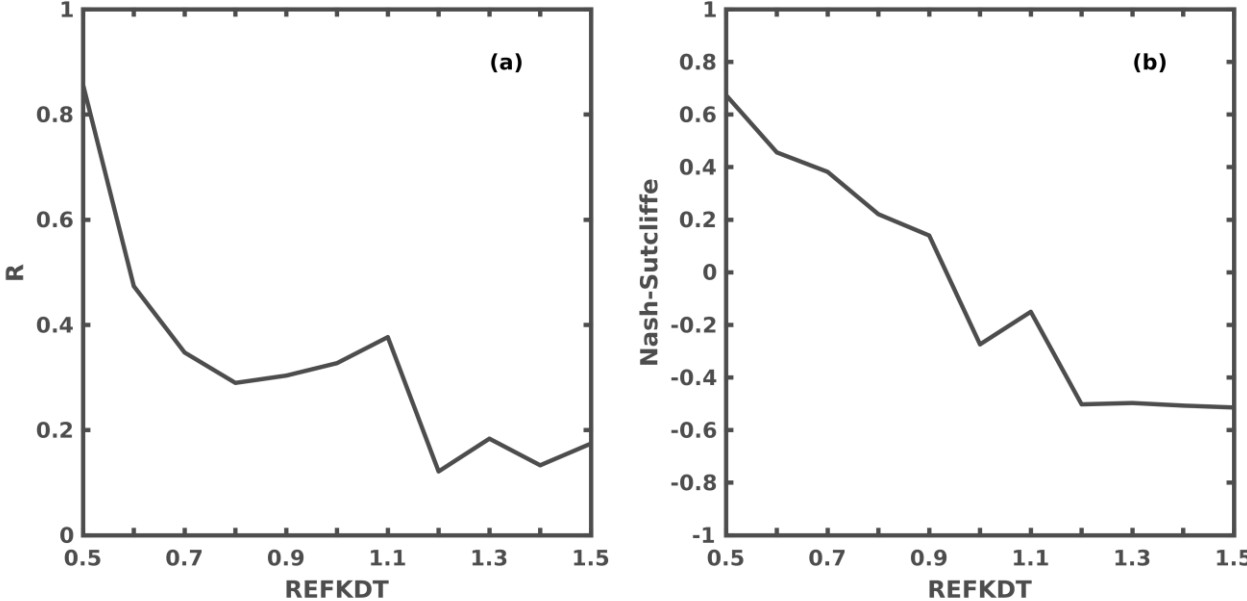

**Figure 5: The temporal distribution of the observed and simulated (a) precipitation and (b) discharge for event #4S and the same variables (c, d) for event #6**


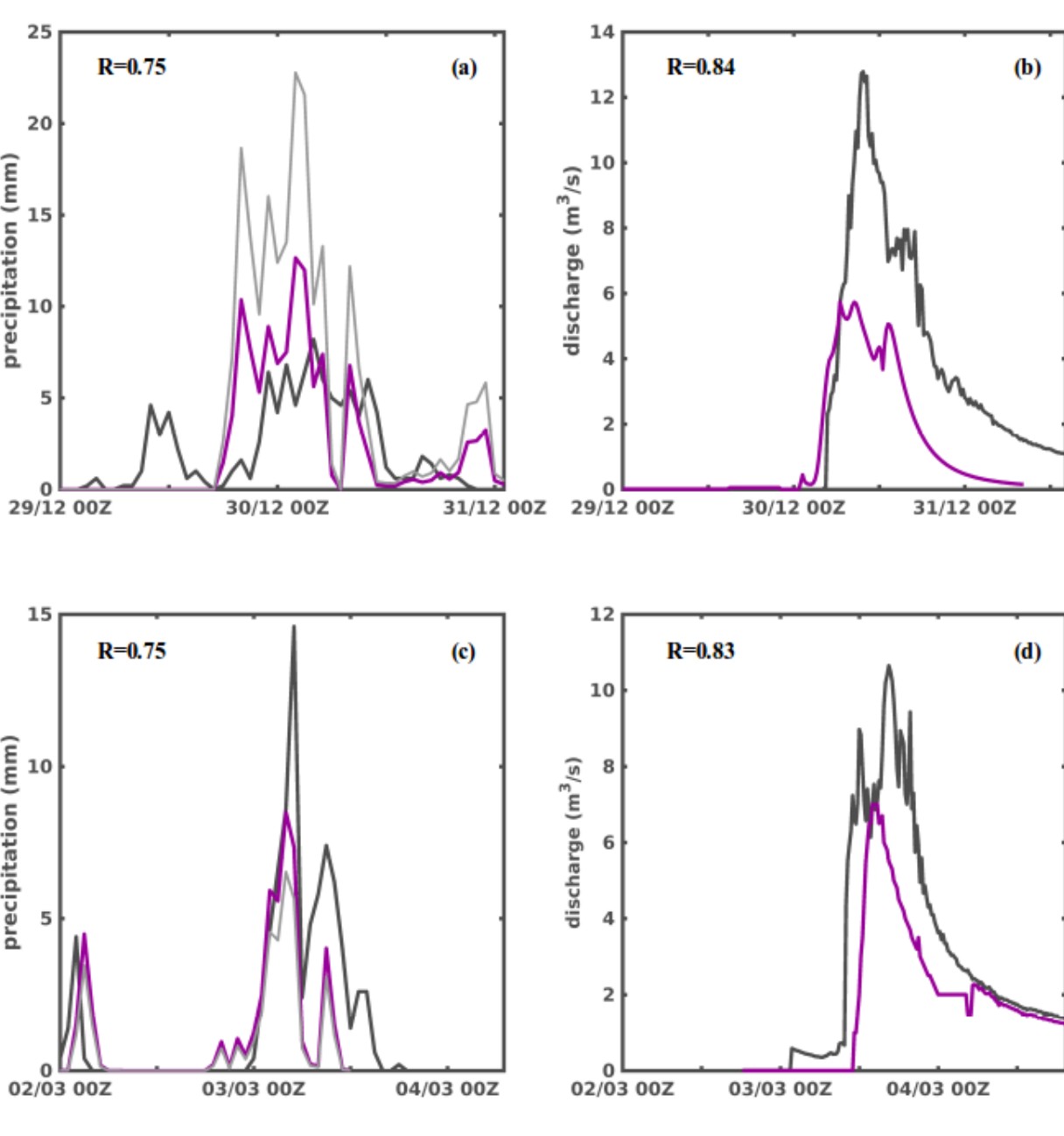

**Figure 6: The temporal evolution of the precipitation in rain gauge station at N.** *Makri for the event #2.*

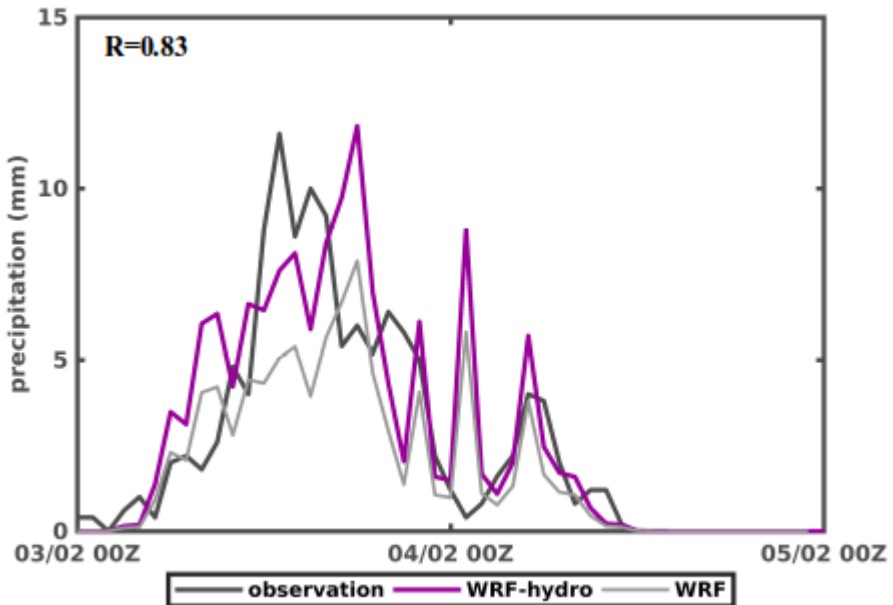


**Figure 7: The evolution of the discharge (observed and simulated) for event #2 for (a) REFKDT, (b) RETDEPRTFAC, (c) OVROUGHRTAC and (d) MannN parameter.**

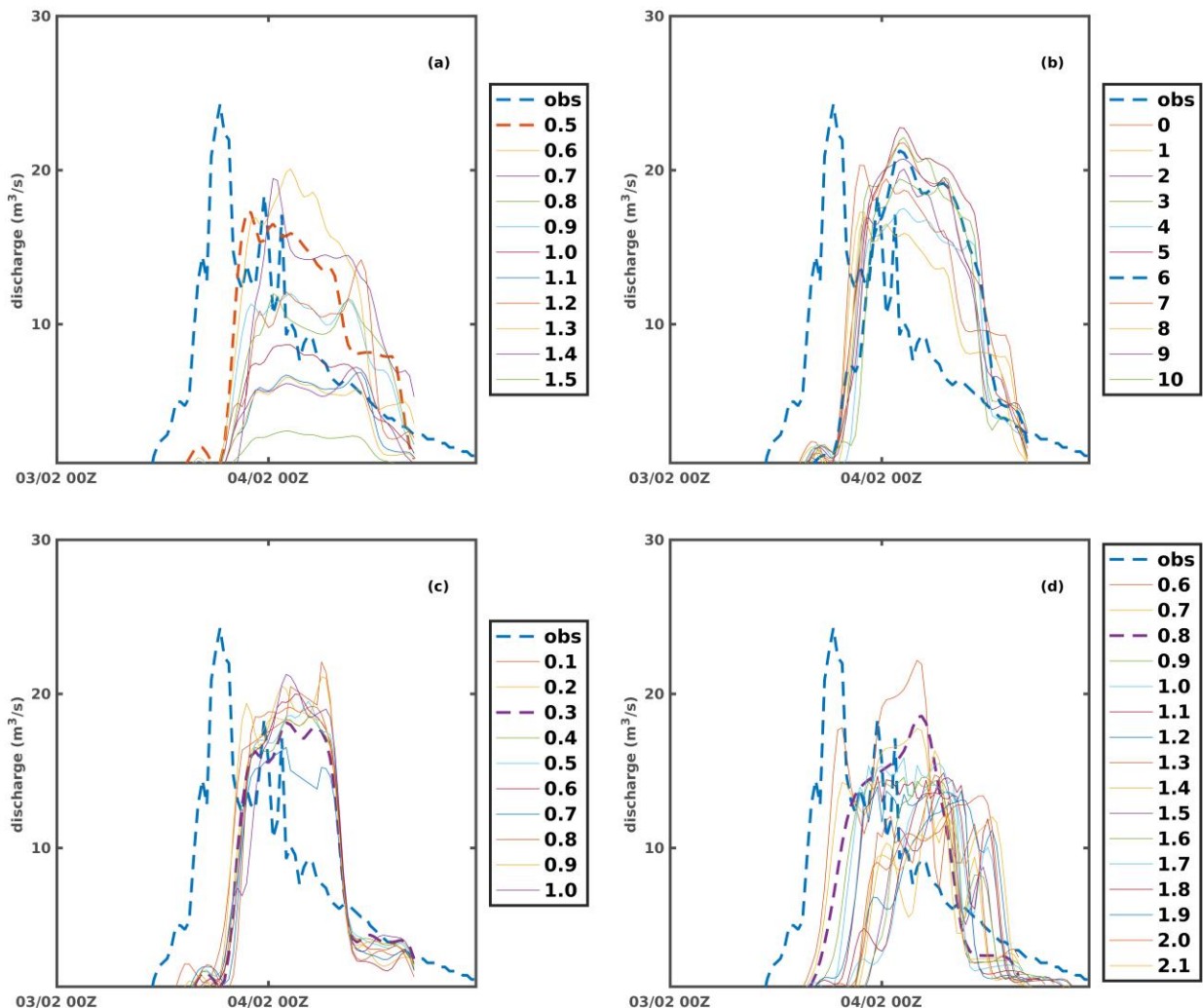

**Figure 8: The temporal distribution of the observed and simulated (a) precipitation and (b) discharge for event #1, and the same variables for event #3 (c, d) and event #4R (e, f)**

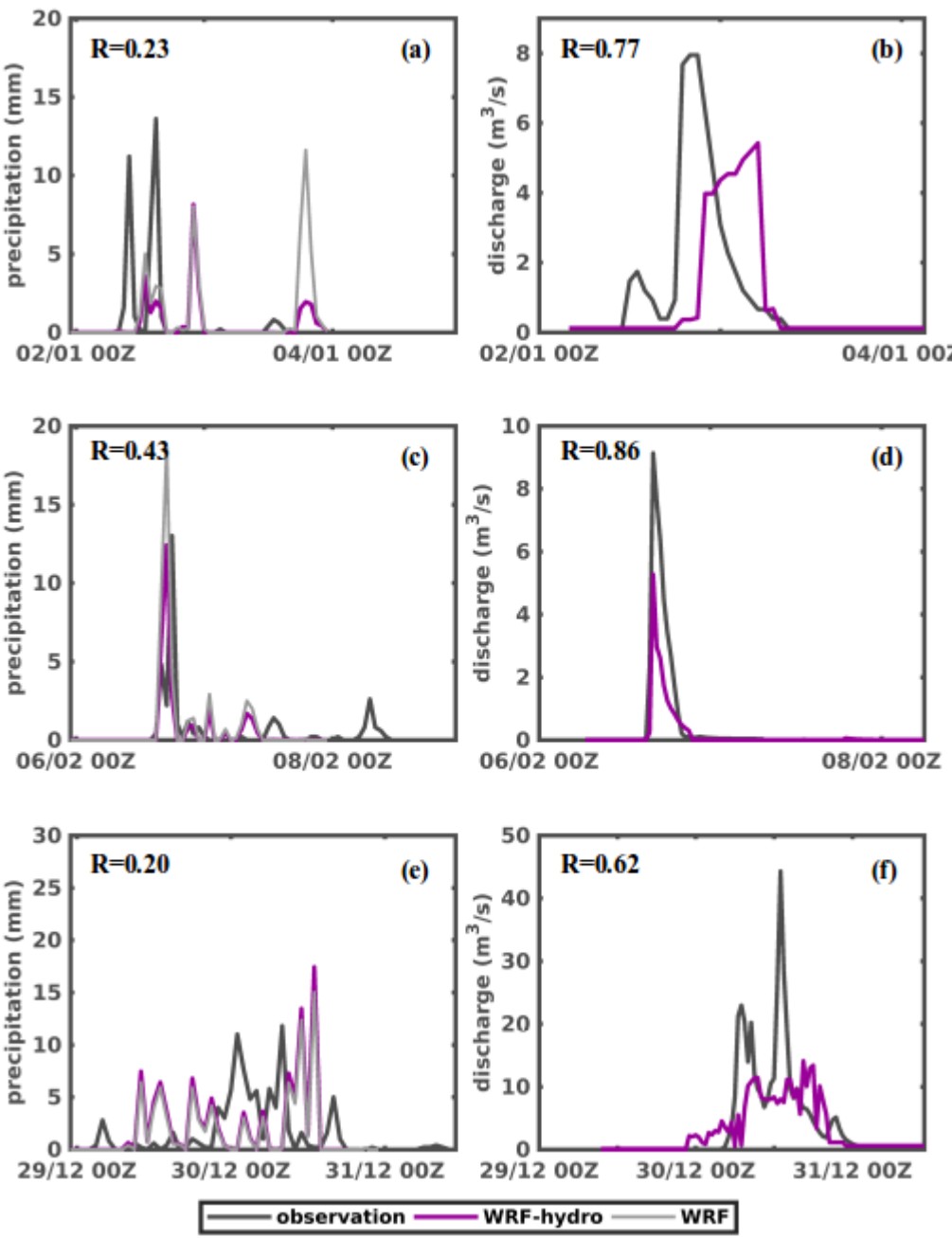

**Figure 9: The difference between observed and simulated (WRF-Hydro and WRF) total amount of precipitation per event for gauge stations of Vilia and N. Makri.**


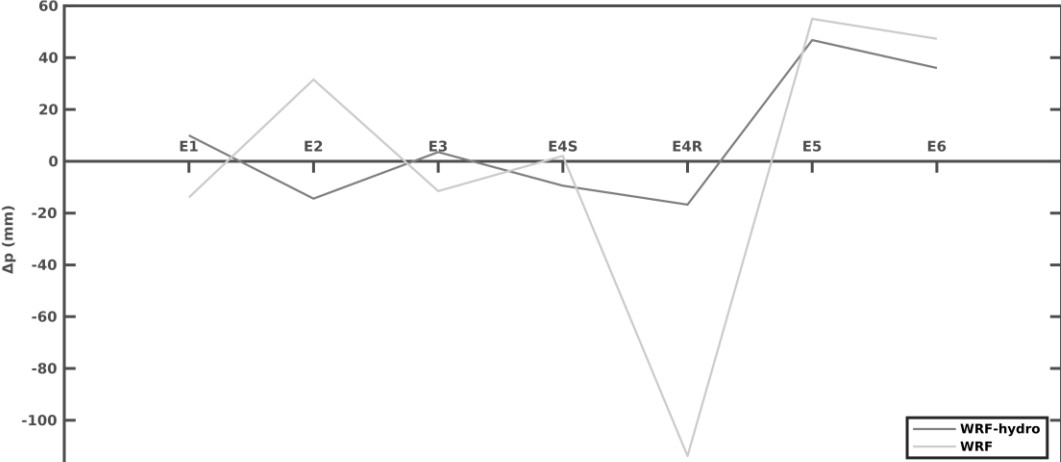