# Peer review of "Implementation of WRF-Hydro at two drainage basins in the region of Attica, Greece for operational flood forecasting"

_Natural Hazards and Earth System Sciences, 2020_

## Referee Comment (RC1) · Anonymous Referee #1 · 2 May 2020

With their paper, Galanaki et al. perform a calibration and validation exercise of the fully coupled WRF-Hydro modelling system over the Attica Region, the most densely populated of Greece, considering 7 high rainfall events from 2011 to 2014. Even though the topic addressed is undoubtedly very interesting (an attempt to perform a complete meteo-hydrological forecast over small catchments in a densely urbanized area), my opinion is that, at least at this stage, the paper does not provide new insights, neither concerning methodology (for which I have some concerns) nor regarding results. The most important novelty, according to authors' words, is that "this outcome is important because WRF-Hydro is implemented under calibration with ground-truth observations for the first time in Greece", but in my opinion, it's not enough (otherwise, any first application of WRF-Hydro around the world should deserve publication). I've some

major comments and several minor comments listed below. My general opinion is that the paper should be strengthened significantly before being ready to publication, even though I acknowledge that some results if presented better and with more details, could be useful and add information to the topic of fully coupled atmospheric-hydrological modelling and its operational application over small catchments. I hope my comments can help with strengthening the study.

Main comments

Introduction: a lot of work made on meteo-hydrological forecasting chains in the Mediterranean area (and in Greece), even using the WRF-Hydro modelling system, has been not considered, but it should. Please find at the end of the review only a partial list of possible references to be considered.

Calibration methods: I've several concerns. Mainly, it's not clear what is the input precipitation for the calibration of the hydrological model (I wonder if the whole fully coupled system was calibrated upon observed discharge). Furthermore, I've doubts about the final choice of the parameters, which not seldom are equal to one of the limits of the range of scaling factors. I also have other doubts for which I ask the authors to refer to my specific comments. Furthermore, I allow myself to suggest authors read the recently accepted paper of Fersch et al. (2020) dealing in the detail with WRF-Hydro calibration issues.

Results: I wonder about the differences between precipitation results with and without fully coupling. Several studies show that for short simulations such as those performed in this study it is very difficult that differences emerge in the precipitation fields due to the differences in soil moisture conditions. Among them, Avolio et al. (2019), which for a case study rather similar to those analyzed by the authors found that correct SST representation is much more impacting. Therefore, more details should be provided by the authors about how they reached their results, and they should try to explain the reasons they got these results.

Furthermore, concerning the presentation of the results themselves, much more details should be given (please refer to specific comments).

Concerning the utility of the study for "operational forecasting purposes", the authors should at least discuss: 1) why they use in their study reanalyses instead of operational GCM forecasts, which makes their study not completely indicative for operational purposes in terms of forecasts performance; 2) what is the additional computational burden of fully coupled simulations and if it's worth it.

Finally, I suggest a general review of the text concerning English grammar and style (some comments, as examples, are provided below).

Specific and minor comments:

L53: Wagner

Fig. 1a: the hydrological features are not clear. I suggest separate panels where the analyzed catchments (including their borders) are represented better. I guess that, given the high urbanization level, land cover is also an important piece of information to highlight. Finally, all the toponyms cited in the text (e.g., Cithaeron mountain range, Halandri's stream, etc.) should be reported in the map

L78: increased concerning what? To the past? What period? Please specify, otherwise, I suggest another term (e.g., high?). Anyway, the sentence looks a bit redundant.

L95: by the Ymittos Mountain

L100: I guess "were provided". This term "provide" is used 4 times in 5 consecutive lines. Probably the text could be revised

L106: I would organize Table 1 from the oldest to the most recent event. Furthermore, I suggest dealing with events #5 and #6 merging them, I guess they depend on the same synoptic situation

L114: "were occurred" not correct

[Figure]

L128: D04

L137: please revise the text

LL139-147: this information should be included in Table 2, possibly along with the corresponding WRF options

L145: it would be useful to explain why the Noah LSM scheme is preferred to the more recent Noah-MP

L157: "The simulation periods for each event are presented in Table 1." Not clear: do the simulations include always the whole days (i.e., from 00:00 to 00:00)? Anyway, what spin-up times were selected?

Section 2.2.2. Even if it is already specified in the title of Section 2.2, I would specify here that WRF-Hydro is used in fully coupled (i.e., two way) mode.

L167: 605/95 = circa 7. So, the disaggregation factor is 7? Please highlight more this feature and explain your choice.

L183: I'm not aware that the stepwise approach is somehow recommended. There are many examples of mixed or automated calibration approaches. Among the others, I suggest a very recent one by Fersch et al. (2020). The cited work of Cuntz et al. refers to Noah-MP, not to WRF-Hydro.

L196: I guess "when a parameter was calibrated"

L196: I understand that there's a kind of hierarchy in parameters calibration, but it's not clear which is the parameter calibrated first and which later

Section 3.1.1: the fundamental information about the initial value of all the calibrated parameters is missing. Furthermore, other information is missing: e.g., what precipitation values were used for the calibration?

L217: the value is at the border of the calibration range. This means that probably the

authors should explore other lower values for REFKDT, relaxing their constraints. The same for RETDEPRTFAC

L219: it's even more unclear what precipitation was used for calibration. I hope observed, not simulated (in Fig. 2 there are two simulated precipitation series)

L224: no displacement would have been necessary if observations were considered.

Figs.2, 5, 6, etc. show both WRF-Hydro and WRF precipitations, but they are not introduced and the difference is not explained in due time into the text.

L245: Figs. 5a and 6a refer to precipitation

L248: time of maximum occurrence?

L251: "time of maximum values": not much better definition than before

Section 3.2: for Rafina catchment, same problems as for the previous calibration procedure (please refer to my comments above)

Section 3.3: what stations are considered? All? Only Vilia and N. Makri? Not clear. If it's only Vilia and N. Makri, how were the other stations shown in fig. 1 used?

L321: Anyah et al.'s work does not regard WRF-Hydro

Conclusions: it looks more like a summary. It should be enriched highlighting the strong points of the study.

References:

Avolio, E., Cavalcanti, O., Furnari, L., Senatore, A., and Mendicino, G.: Brief communication: Preliminary hydro-meteorological analysis of the flash flood of 20 August 2018 in Raganello Gorge, southern Italy, Nat. Hazards Earth Syst. Sci., 19, 1619–1627, https://doi.org/10.5194/nhess-19-1619-2019, 2019.

Fersch, B., Senatore, A., Adler, B., Arnault, J., Mauder, M., Schneider, K., Völksch, I., and Kunstmann, H.: High-resolution fully-coupled atmospheric–hydrological modeling:

a cross-compartment regional water and energy cycle evaluation, Hydrol. Earth Syst. Sci. Discuss., https://doi.org/10.5194/hess-2019-478, in review, 2019.

Papaioannou, G.; Varlas, G.; Terti, G.; Papadopoulos, A.; Loukas, A.; Panagopoulos, Y.; Dimitriou, E. Flood Inundation Mapping at Ungauged Basins Using Coupled Hydrometeorological–Hydraulic Modelling: The Catastrophic Case of the 2006 Flash Flood in Volos City, Greece. Water 2019, 11, 2328.

Senatore, A., Furnari, L., and Mendicino, G.: Impact of high-resolution sea surface temperature representation on the forecast of small Mediterranean catchments' hydrological responses to heavy precipitation, Hydrol. Earth Syst. Sci., 24, 269–291, https://doi.org/10.5194/hess-24-269-2020, 2020.

Varlas, G.; Anagnostou, M.N.; Spyrou, C.; Papadopoulos, A.; Kalogiros, J.; Mentzafou, A.; Michaelides, S.; Baltas, E.; Karymbalis, E.; Katsafados, P. A Multi-Platform Hydrometeorological Analysis of the Flash Flood Event of 15 November 2017 in Attica, Greece. Remote Sens. 2019, 11, 45.

---

## Referee Comment (RC2) · Anonymous Referee #2 · 27 Sep 2020

**General comment**

The coupling of land and atmospheric processes and evaluating the impact on the forecast skill compared to atmosphere-only modeling is an important topic for the community and particularly NHESS readers. The manuscript aims to (1) to investigate the ability of WRF-Hydro to simulate selected cases of flood occurrence in the area of Attica (Greece) and (2) to study the influence of land-atmosphere interactions on the improvement of precipitation forecasting. While the first objective is an important effort towards local operational flood forecasting, the second objective would be the main source of novelty and new insights for the scientific community. However, the current version of the manuscript does not thoroughly address this objective and fails to diagnose the physical mechanism explaining the reported improvement from the coupling. My suggestion would be a re-submission after the authors make the below major improvement which may/may not alter the main conclusions of the study.

**Major comments**

Comment 1:

In order to take the full advantage of the WRF-Hydro system, diagnoses of the feedback processes/mechanisms controlling the water cycle (e.g. runoff, penetration, evaporative fraction, water vapor flux) should be conducted. Such diagnoses may lead to valuable generic outcome that could benefit the research community. The primary mechanism to diagnose is the soil moisture-precipitation feedback loop (El Tahir et al., 1998) and the evolution of surface fluxes during the simulations (uncoupled vs. coupled) – see for example the recent works of Kumar et al. (2020) and Wehbe et al (2019). It is strongly recommended that such diagnoses are explored to confirm speculative statements, such as that mentioned in Line 302: "The improved simulation of the soil moisture affects the computation of the sensible and latent heat fluxes, which influence humidity and temperature in the lower atmosphere and consequently precipitation. Therefore, the physical process of the coupling of land-atmosphere is expected to improve the forecast skill of precipitation".

Comment 2:

Please specify if a two-way or one-way grid nesting was employed. This is a crucial point.

If a one-way grid nesting was used, the authors have to make sure that domains 1, 2 and 3 are identical in both WRF and WRF-Hydro simulations. This may not be the case if the authors used two different executables, one for WRF and the other for WRF-Hydro. If domains 1, 2 and 3 in the WRF and WRF-Hydro simulations are different, then it can be argued that the differences obtained in domain 4 are not due to the consideration of lateral hydrological processes, but to different large-scale forcing. In this case the main conclusion of the paper has to be revised.

If a two-way grid nesting was used, then the above effect is masked by the feedbacks from domain 4, which are unlikely to be exactly the same between the WRF and WRF-Hydro simulations. Still, the fact that domain 1, 2 and 3 would be different in this case would not be necessarily due to the feedbacks from the resolved lateral water flow in domain 4, but simply internal atmospheric variability. The authors are very quick in concluding that the improved precipitation in the WRF-Hydro simulation is due to the coupling with lateral terrestrial hydrological processes, which is then taken for granted through the rest

of the manuscript. But in my opinion, this improvement would rather be due to atmospheric internal variability, which is a well-known limitation of regional atmospheric models (e.g. Rassmussen et al. 2012).

So in any case the authors have to provide an estimation of internal atmospheric variability, in order to prove that the claimed improvement in modeled precipitation with WRF-Hydro is not the result of a random realization of the considered atmospheric situation. In other words, the authors have to provide an ensemble and assess the robustness of a potential improvement with WRF-Hydro. The ensemble could be generated, for example, by disturbing the initial condition, or by using the GEFS ensemble forecast runs. This ensemble could simply be generated, for example, by adding random perturbation in the soil moisture initial condition, or whatever prognostic variable.

Comment 3:

Why was event #2 selected for the calibration among the other events? Please add more details on the structure/scale of these events – were they all microscale, mesoscale or synoptic situations? This has severe implications on the robustness of the conclusions which may be governed by the microphysics options rather than the WRF-Hydro coupling. The authors select the WSM6 microphysics scheme without providing any justification. Are their previous sensitivity studies done for Greece or the surrounding region to support this selection and its relevance to the simulated storm scale(s)?

**Minor comments/corrections**

Line 8 (abstract): This study presents an integrated modeling approach for simulating flood events.

Line 12: Remove "on the improvement of"

Line 14: carried out with "the" WRF-Hydro model. There should also be mention of the comparison with WRF-only (standalone/uncoupled) runs.

Line 26: …especially "in its capital, Athens," flooding events…

Line 51: revise to "WRF-Hydro is a recently developed coupled hydrometeorological system that has been used for numerous research applications

Line 61: remove "the" before 36%

Line 75: add "the" before Cithaeron

Line 86: revise to "In the current study, we focus on two…"

Line 89: replace "intense" with "increasing" before urbanization

Line 100-103: capitalize "H" in "WRF-hydro" and correct the sentence structure.

Line 106: "Namely" is used incorrectly here

Line 113: add of: "...the whole **of** Greece…"

Line 137: add for "…of the area **for** better simulation…"

Line 140: please justify the selection of WSM6 MP scheme for the study domain. Are their sensitivity studies done for Greece or the surrounding region to support this selection?

Line 145: please justify the selection of the NOAH LSM instead of the NOAH-MP LSM (also comment on the selection of the MYJ PBL scheme vs. other schemes).

Line 218: Use either the long dash (–) or short dash (-) concisely for the term Nash-Sutcliffe

Figures:

Merge figures 5 and 6 using subplots and add error metrics on each subplot

Merge figures 9, 10 and 11 using subplots and add error metrics on each subplot

**References:**

- Eltahir, E. A. (1998). A soil moisture–rainfall feedback mechanism: 1. Theory and observations. *Water resources research*, *34*(4), 765-776.
- Kumar, S., Newman, M., Lawrence, D. M., Lo, M. H., Akula, S., Lan, C. W., ... & Lombardozzi, D. (2020). The GLACE-Hydrology Experiment: Effects of Land–Atmosphere Coupling on Soil Moisture Variability and Predictability. *Journal of Climate*, *33*(15), 6511-6529.
- Wehbe, Y., Temimi, M., Weston, M., Chaouch, N., Branch, O., Schwitalla, T., ... & Al Mandous, A. (2019). Analysis of an extreme weather event in a hyper-arid region using WRF-Hydro coupling, station, and satellite data. *Natural Hazards & Earth System Sciences*, *19*(6).
- Rasmussen, S. H., Christensen, J. H., Drews, M., Gochis, D. J., & Refsgaard, J. C. (2012). Spatial-scale characteristics of precipitation simulated by regional climate models and the implications for hydrological modeling. *Journal of Hydrometeorology*, *13*(6), 1817-1835.

---

## Author Comment (AC1) · 7 Dec 2020

With their paper, Galanaki et al. perform a calibration and validation exercise of the fully coupled WRF-Hydro modelling system over the Attica Region, the most densely populated of Greece, considering 7 high rainfall events from 2011 to 2014. Even though the topic addressed is undoubtedly very interesting (an attempt to perform a complete meteo-hydrological forecast over small catchments in a densely urbanized area), my opinion is that, at least at this stage, the paper does not provide new insights, neither concerning methodology (for which I have some concerns) nor regarding results. The most important novelty, according to authors' words, is that "this outcome is important because WRF-Hydro is implemented under calibration with ground-truth observations for the first time in Greece", but in my opinion, it's not enough (otherwise, any first application of WRF-Hydro around the world should deserve publication). I've some major comments and several minor comments listed below. My general opinion is that the paper should be strengthened significantly before being ready to publication, even though I acknowledge that some results if presented better and with more details, could be useful and add information to the topic of fully coupled atmospheric-hydrological modelling and its operational application over small catchments. I hope my comments can help with strengthening the study.

Main comments

Introduction: a lot of work made on meteo-hydrological forecasting chains in the Mediterranean area (and in Greece), even using the WRF-Hydro modelling system, has been not considered, but it should. Please find at the end of the review only a partial list of possible references to be considered.

More studies related to numerical hydrometeorological research has been cited in the Introduction Sections (lines 67-71).

**Lines 67-71:**

"…The WRF-Hydro model has been used in numerous flood-related research applications (Senatore et al., 2020; Papaioannou et al., 2019; Varlas et al., 2019; Avolio et al., 2019; Lin et al., 2018; Silver et al., 2017; Xiang et al. 2017; Arnault et al., 2016; Givati et al., 2016; Wagner et al., 2016; Senatore et al., 2015; Yucel et al., 2015) and for operational flood forecasting in the United States (Krajewski et al., 2017; NOAA, 2016) and Israel (Givati and Sapir, 2014)."

Calibration methods: I've several concerns. Mainly, it's not clear what is the input precipitation for the calibration of the hydrological model (I wonder if the whole fully coupled system was calibrated upon observed discharge). Furthermore, I've doubts about the final choice of the parameters, which

not seldom are equal to one of the limits of the range of scaling factors. I also have other doubts for which I ask the authors to refer to my specific comments. Furthermore, I allow myself to suggest authors read the recently accepted paper of Fersch et al. (2020) dealing in the detail with WRF-Hydro calibration issues.

**Concerning the precipitation:**

The calibration of the WRF-Hydro model was performed based on the WRF atmospheric forcing, including the precipitation fields. Several preliminary tests have been performed concerning the WRF model configuration (spin-up, physics parameterization; lines 175-177 and 199-204) in order to achieve the most accurate representation of the observed precipitation which is of great importance for simulating the corresponding observed discharge. Corrections have been applied to the manuscript to clarify the above (lines 261-263).

It is worth mentioning that previous studies calibrated the WRF-Hydro model following the same approach of forcing the model with WRF data (e.g., Li et al. 2020; Liu et al., 2020; Li et al. 2017; Silver et al., 2017).

**Lines 261-263:**

"…The calibration of the WRF-Hydro model was performed using the WRF atmospheric forcing, including the precipitation fields, following the same approach of forcing the model with WRF data from previous studies (e.g. Li et al. 2020; Liu et al., 2020; Li et al. 2017)."

**Concerning the calibrated parameters:**

The reviewer is right. The manuscript was modified to highlight this fact (lines 300-301 and 338-339).

**Lines 300-301:**

"…It should be noted that the optimal parameters for REFKDT and RETDEPRTFAC hit the lower and calibration limit, respectively. Relaxing their constraints may result to better calibrations results."

**Lines 338-339:**

"…As in the case of Sarantapotamos, the optimum value for REFKDT reaches the lower calibration limit indicating that changing the calibration limit may let to better result."

Results: I wonder about the differences between precipitation results with and without fully coupling. Several studies show that for short simulations such as those performed in this study it is very difficult that differences emerge in the precipitation fields due to the differences in soil

moisture conditions. Among them, Avolio et al. (2019), which for a case study rather similar to those analyzed by the authors found that correct SST representation is much more impacting. Therefore, more details should be provided by the authors about how they reached their results, and they should try to explain the reasons they got these results.

The differences in the simulated precipitation between WRF-only and WRF-Hydro models have been addressed by examining the soil moisture and latent heat flux before the initiation of the precipitation for each event. Slight differences between the average values of the aforementioned parameters were found, which may affect the resulted precipitation. The authors are aware that this outcome is an indication, as highlighted in the manuscript, and that the effects of soil moisture on precipitation fields are more evident in long-term simulations, when the land surface variables read a steady state (e.g., Senatore et al., 2015). For this, they intend to perform an in-depth analysis for assessing the model's surface energy budget in a follow-up study.
The manuscript was modified to clarify the above (lines 388-399 and 436-447)

**Lines 388-399:**

"…Table 7 shows the basin average soil moisture (at the $1^{st}$ level) and latent heat flux simulated by the WRF-Hydro and WRF-only models, at the time before the beginning of the examined storms events. As can be seen the soil moisture differences between the models range from 0.005 to 0.027 $m^3$ $m^{-3}$ and latent heat flux differences span from 0.038 to 16.862 $W/m^2$. These differences simulated by the two models provides an indication that the most accurate replication of the observed precipitation provided by the WRF-Hydro model compared to the WRF-only model is related to the physical process associated with the coupling of land-atmosphere and hydrological routing in the WRF-Hydro model. In particular, WRF-Hydro, affects the soil moisture content, due to the computation of the lateral redistribution and re-infiltration of the water (Gochis et al., 2013), which in turn influences the computation of the sensible and latent heat fluxes. These fluxes are associated with humidity and temperature in the lower atmosphere and consequently precipitation (Seneviratne et al., 2010). However, it should be noted that the effects of soil moisture on precipitation fields are more evident and valid in long-term simulations when the land surface variables reach a steady state (Fersch et al., 2020; Senatore et al., 2015)."

**Lines 436-447:**

"A preliminary analysis of key water budget components indicated that the precipitation simulation improvement provided by the WRF-Hydro system may be related to the feedback of the terrestrial hydrology parameterization on the modeled atmosphere. A follow up study could focus on the further investigation of impact of the more detailed representation of the interaction between the land surface and hydrology processes to the surface energy budget under the WRF-Hydro coupling scheme by applying long-term simulations and validated the results against ground-based or satellite observation, considering limitations arising from internal model variability (Bassett et al., 2020) and domain size (Fersch et al, 2020; Arnault et al., 2018). Also, the incorporation of the SST update into the model will be considered as previous studies shown a positive feedback to simulations (Avolio et al., 2019; Senatore et al., 2015).

Even though a more detailed analysis is required to explore the sensitivity of the simulated precipitation to the coupling between hydrological and land-atmosphere processes, the current study demonstrates that the coupled WRF-Hydro model has the potential to enhance precipitation forecast skill for operational flood predictions."

Furthermore, concerning the presentation of the results themselves, much more details should be given (please refer to specific comments).

Please find the author's responses in the specific comments.

Concerning the utility of the study for "operational forecasting purposes", the authors should at least discuss: 1) why they use in their study reanalyses instead of operational GCM forecasts, which makes their study not completely indicative for operational purposes in terms of forecasts performance; 2) what is the additional computational burden of fully coupled simulations and if it's worth it.

1) Unfortunately, the on-line availability of the GFS forecasts is limited for historical periods as the studied one (2011- 2014). GFS initialization data could be ordered for the investigated events but at a coarse spatial resolution (0.5°x0.5°), which was not consider adequate for forcing the WRF simulations having a coarse domain (do1) resolution of 18 km. For this, the ERA5 reanalysis data were preferred over the GFS operational forecasts in this study. Concerning the ECMWF IFS forecasts, unfortunately, their availability is restricted to National meteorological services or users with a special paid contract. The manuscript has been modified accordingly (lines 195-198)

**Lines 195-198:**

"…It should be noted that the use of ERA5 reanalysis data was preferred instead of the operational GFS data, as the on-line availability of the GFS forecasts is limited for historical periods. GFS initialization data could be ordered for the investigated events but at a high spatial resolution of 0.5° × 0.5°, which was not considered adequate for forcing the WRF simulations having a coarse domain (do1) resolution of 18 km."

2) The manuscript has been modified accordingly to address the computation burden of fully coupled simulations (lines 448-454)

**Lines 448-453:**

"…For an operational point of view, the application of a coupled WRF-Hydro model to exploit its beneficial impact in simulating precipitation is partially limited due to the additional computational time needed for the execution of the WRF-Hydro model. In particular, in our case, a three day coupled WRF-Hydro forecast considering a prior 12 hours spin up under the investigated configuration requires x1.35 time compares to WRF-only implementation in 140

computing nodes. It should be noted that the extra computational time depends on the WRF-Hydro configuration and the computing resources, in which the model is applied."

Finally, I suggest a general review of the text concerning English grammar and style (some comments, as examples, are provided below).

Revisions concerning the English were made throughout the whole manuscript.

Specific and minor comments:

L53: Wagner

Changed accordingly.

Fig. 1a: the hydrological features are not clear. I suggest separate panels where the analyzed catchments (including their borders) are represented better. I guess that, given the high urbanization level, land cover is also an important piece of information to highlight. Finally, all the toponyms cited in the text (e.g., Cithaeron mountain range, Halandri's stream, etc.) should be reported in the map

Fig. 1 was updated accordingly.

L78: increased concerning what? To the past? What period? Please specify, otherwise, I suggest another term (e.g., high?). Anyway, the sentence looks a bit redundant.

The sentence was corrected.

L95: by the Ymittos Mountain

Corrected.

L100: I guess "were provided". This term "provide" is used 4 times in 5 consecutive lines. Probably the text could be revised

Lines 117-126 were modified to address this issue.

L106: I would organize Table 1 from the oldest to the most recent event. Furthermore, I suggest dealing with events #5 and #6 merging them, I guess they depend on the same synoptic situation.

Table has been organized according to the reviewer's suggestion. The old events #1 and #7 have been merged (new event #4), while the old events #5 (new event #2) and #6 (new event #3) were kept separately as they refer to different dates, and, consequently, they are characterized by different atmospheric conditions (lines 127-154).

L114: "were occurred" not correct

Changed to "were reported"

L128: D04

Corrected.

L137: please revise the text

The text was revised.

LL139-147: this information should be included in Table 2, possibly along with the corresponding WRF options

Table 2 was updated accordingly.

L145: it would be useful to explain why the Noah LSM scheme is preferred to the more recent Noah-MP

The manuscript was modified to justify the use of the Noah LSM (lines 185-189).

**Lines 185-189:**

"...Noah-MP introduces multiple options and tunable parameters to simulate the land surface processes. However, the default values of these options and parameters are not suitable for every study area (e.g. Giannaros et al., 2019). In contrast, the Noah LSM has been tested and applied successfully in several studies focusing in Greece (e.g. Varlas et al., 2019; Papaioannou et al., 2019; Giannaros et al., 2020)."

L157: "The simulation periods for each event are presented in Table 1." Not clear: do the simulations include always the whole days (i.e., from 00:00 to 00:00)? Anyway, what spin-up times were selected?

The spin-up time and the exact time of the simulations' start and end are now included in Table 1.

Section 2.2.2. Even if it is already specified in the title of Section 2.2, I would specify here that WRF-Hydro is used in fully coupled (i.e., two way) mode.

The manuscript was changed accordingly.

L167: 605/95 = circa 7. So, the disaggregation factor is 7? Please highlight more this feature and explain your choice.

More information was added concerning the choice of disaggregation factor (lines 215-219).

**Lines 215-219:**

"…The catchments' routing grids were computed based on SRTM 90 m topography data using the WRF-Hydro GIS pre-processing toolkit. In order to exploit this high-resolution input dataset, avoiding interpolation to a coarser grid (Verri et al., 2017; Gochis and Chen, 2003), a ~95 m spatial resolution WRF-Hydro domain was configured over the WRF innermost domain. Thus, the ratio between the high-resolution terrain routing grid and the WRF land surface model (aggregation factor; AGGFACTRT) was set to 7."

L183: I'm not aware that the stepwise approach is somehow recommended. There are many examples of mixed or automated calibration approaches. Among the others, I suggest a very recent one by Fersch et al. (2020). The cited work of Cuntz et al. refers to Noah-MP, not to WRF-Hydro.

The reviewer is right. The manuscript was modified accordingly.

L196: I guess "when a parameter was calibrated"

Corrected.

L196: I understand that there's a kind of hierarchy in parameters calibration, but it's not clear which is the parameter calibrated first and which later

The manuscript was modified to clarify this issue (lines 252-253).

**Lines 252-253:**

"…Thus, the parameters were calibrated in the following order: REFKDT, RETDEPRTFAC, OVROUGHRTAC and MannN."

Section 3.1.1: the fundamental information about the initial value of all the calibrated parameters is missing. Furthermore, other information is missing: e.g., what precipitation values were used for the calibration?

Table 3 has been updated to include the default values of the calibrated parameters.
Concerning precipitation, please refer to the main comment concerning calibrated methods.

L217: the value is at the border of the calibration range. This means that probably the authors should explore other lower values for REFKDT, relaxing their constraints. The same for RETDEPRTFAC

Please refer to the main comment concerning calibrated methods.

L219: it's even more unclear what precipitation was used for calibration. I hope observed, not simulated (in Fig. 2 there are two simulated precipitation series)
L224: no displacement would have been necessary if observations were considered.

Please refer to the main comment concerning calibrated methods.

Figs.2, 5, 6, etc. show both WRF-Hydro and WRF precipitations, but they are not introduced and the difference is not explained in due time into the text.

The authors consider essential the fields of observed and simulated (WRF-Hydro) temporal evolution of precipitation to be in the same subplot with the observed and simulated temporal evolution of discharge. Indeed, the discussion concerning the simulated temporal evolution of precipitation from WRF-only simulations is introduced at the last section of the results. We could extract the field of precipitation from WRF-only simulations from the existed figures and reproduce the same figures for the precipitation at the sector 3.3, but we consider that it will be confusing to show these figures twice.

L245: Figs. 5a and 6a refer to precipitation

Corrected.

L248: time of maximum occurrence?

Corrected.

L251: "time of maximum values": not much better definition than before

Corrected.

Section 3.2: for Rafina catchment, same problems as for the previous calibration procedure (please refer to my comments above)

The corresponding corrections were applied in the manuscript

Section 3.3: what stations are considered? All? Only Vilia and N. Makri? Not clear. If it's only Vilia and N. Makri, how were the other stations shown in fig. 1 used?

The analysis was performed using only the stations of Vilia and N. Makri. Corrections have been applied in the manuscript to clarify this fact.
The remaining stations in the old Fig.1 have been utilized in the initial sensitivity tests for finding the best configuration of WRF, the result of which are not included in the manuscript. Fig. 1 was updated to avoid any misconceptions.

L321: Anyah et al.'s work does not regard WRF-Hydro

Removed.

Conclusions: it looks like a summary. It should be enriched highlighting the strong points of the study

This part of the manuscript were modified. We added additional information related to the water budget analysis and the computational burden of the hydrological analysis.

---

## Author Comment (AC2) · 7 Dec 2020

**General comment**

The coupling of land and atmospheric processes and evaluating the impact on the forecast skill compared to atmosphere-only modeling is an important topic for the community and particularly NHESS readers. The manuscript aims to (1) to investigate the ability of WRF-Hydro to simulate selected cases of flood occurrence in the area of Attica (Greece) and (2) to study the influence of land-atmosphere interactions on the improvement of precipitation forecasting. While the first objective is an important effort towards local operational flood forecasting, the second objective would be the main source of novelty and new insights for the scientific community. However, the current version of the manuscript does not thoroughly address this objective and fails to diagnose the physical mechanism explaining the reported improvement from the coupling. My suggestion would be a re-submission after the authors make the below major improvement which may/may not alter the main conclusions of the study.

**Major comments**

**Comment 1:**

In order to take the full advantage of the WRF-Hydro system, diagnoses of the feedback processes /mechanisms controlling the water cycle (e.g. runoff, penetration, evaporative fraction, water vapor flux) should be conducted. Such diagnoses may lead to valuable generic outcome that could benefit the research community. The primary mechanism to diagnose is the soil moisture-precipitation feedback loop (El Tahir et al., 1998) and the evolution of surface fluxes during the simulations (uncoupled vs. coupled) – see for example the recent works of Kumar et al. (2020) and Wehbe et al (2019). It is strongly recommended that such diagnoses are explored to confirm speculative statements, such as that mentioned in Line 302: "The improved simulation of the soil moisture affects the computation of the sensible and latent heat fluxes, which influence humidity and temperature in the lower atmosphere and consequently precipitation. Therefore, the physical process of the coupling of land-atmosphere is expected to improve the forecast skill of precipitation".

**Comment 2:**

Please specify if a two-way or one-way grid nesting was employed. This is a crucial point.

If a one-way grid nesting was used, the authors have to make sure that domains 1, 2 and 3 are identical in both WRF and WRF-Hydro simulations. This may not be the case if the authors used two different executables, one for WRF and the other for WRF-Hydro. If domains 1, 2 and 3 in the WRF and WRF-Hydro simulations are different, then it can be argued that the differences obtained in domain 4 are not due to the consideration of lateral hydrological processes, but to different large-scale forcing. In this case the main conclusion of the paper has to be revised.

If a two-way grid nesting was used, then the above effect is masked by the feedbacks from domain 4, which are unlikely to be exactly the same between the WRF and WRF-Hydro simulations. Still, the fact that domain 1, 2 and 3 would be different in this case would not be necessarily due to the feedbacks from the resolved lateral water flow in domain 4, but simply internal atmospheric variability. The authors are very quick in concluding that the improved precipitation in the WRF-Hydro simulation is due to the coupling with lateral terrestrial hydrological processes, which is then taken for granted through the rest of the manuscript. But in my opinion, this improvement would rather be due to atmospheric internal variability, which is a well-known limitation of regional atmospheric models (e.g. Rassmussen et al. 2012).

So in any case the authors have to provide an estimation of internal atmospheric variability, in order to prove that the claimed improvement in modeled precipitation with WRF-Hydro is not the result of a random realization of the considered atmospheric situation. In other words, the authors have to provide an ensemble and assess the robustness of a potential improvement with WRF-Hydro. The ensemble could be generated, for example, by disturbing the initial condition, or by using the GEFS

ensemble forecast runs. This ensemble could simply be generated, for example, by adding random perturbation in the soil moisture initial condition, or whatever prognostic variable.

**Concerning the reviewer's suggestions in main comments 1 and 2:**
Indeed, taking full advantage of a two-way coupled hydrometeorological model requires assessing its ability to improve the physical realism concerning land-atmosphere and hydrological interactions, and their impact on precipitation. Such an assessment is more relevant to long-term simulations, when the land surface variables reach a steady state and affect more evidently the precipitation formation (e.g., Senatore et al., 2015). Also, the authors acknowledge that internal model variability (IVM) is an important issue concerning regional atmospheric models (e.g., Bassett et al., 2020). However, both the detailed analysis of the model's water and energy budget and the investigation of uncertainties arising from IVM are out of the scope of the preset study.
The current study aims principally on assessing the capability of the coupled WRF-Hydro model as an operational short-term flood forecasting system, as given the susceptibility of the study area (Attica) to flooding, which is sufficiently described in the introduction, the development of such an operational tool is considered of great importance. In this framework, the study also investigated the impact of applying a coupled hydrometeorological model on the precipitation forecast skill. The results showed that the coupled WRF-Hydro model has the potential to improve the precipitation forecast accuracy, which is essential for flood forecasting purposes. Following the reviewer's suggestion, a **preliminary analysis was added to the manuscript regarding key water budget components**, indicating that the precipitation simulation improvement provided by the WRF-Hydro system may be related to the feedback of the terrestrial hydrology parameterization on the modeled atmosphere. The authors acknowledge that this outcome is just an indication and that a more detailed analysis is required to confirm this. Recognizing the importance of such an in-depth analysis, the authors intend to perform it in the future as a follow-up study, considering limitations arising from IVM.
The manuscript was modified to clarify the above (lines 387-399, 436-447), as well as the nesting approach applied for the simulations (lines 158-162)

**Lines 387-399:**
"…Table 7 shows the basin average soil moisture (at the 1st level) and latent heat flux simulated by the WRF-Hydro and WRF-only models, at the time before the beginning of the examined storms events. As can be seen the soil moisture differences between the models range from 0.005 to 0.0269 $m^3$ $m^{-3}$ and latent differences span from 0.0376 to 16.8621 $W/m^2$. These differences simulated by the two models provides an indication that the most accurate replication of the observed precipitation provided by the WRF-Hydro model compared to the WRF-only model is related to the physical process associated with the coupling of land-atmosphere and hydrological routing in the WRF-Hydro model. In particular, WRF-Hydro, affects the soil moisture content, due to the computation of the lateral redistribution and re-infiltration of the water (Gochis et al., 2013), which in turn influences the computation of the sensible and latent heat fluxes. These fluxes are associated with humidity and temperature in the lower atmosphere and consequently precipitation (Seneviratne et al., 2010). However, it should be noted that the effects of soil moisture on precipitation fields are more evident and valid in long-term simulations when the land surface variables reach a steady state (Fersch et al., 2020; Senatore et al., 2015)."

**Lines 436-447:**
"A preliminary analysis of key water budget components indicated that the precipitation simulation improvement provided by the WRF-Hydro system may related to the feedback of the terrestrial hydrology parameterization on the modeled atmosphere. A follow up study could focus on the further investigation of impact of the more detailed representation of the interaction between the land surface and hydrology processes to the surface energy budget under the WRF-Hydro coupling scheme by applying long-term simulations and validated the results against ground-based or

satellite observation, considering limitations arising from internal model variability (Bassett et al., 2020) and domain size (Fersch et al, 2020; Arnault et al., 2018). Also, the incorporation of the SST update into the model will be consider as previous studies shown a positive feedback to simulations (Avolio et al., 2019; Senatore et al., 2015). Even though a more detailed analysis is required to explore the sensitivity of the simulated precipitation to the coupling between hydrological and land-atmosphere processes, the current study demonstrates that the coupled WRF-Hydro model has the potential to enhance precipitation forecast skill for operational flood predictions."

**Lines 158-162:**
"…The Advanced Research Weather Research and Forecasting model Version 3.9.1.1 was used in this study (Skamarock et al., 2008) for the land-atmosphere simulations which were carried out using four two-way nested grids (Fig. 1b): d01, d02, d03 d04 with 18 km (325 × 285 grid points), 6 km (685 × 337 grid points), 2 km (538 × 499 grid points) and 667 m (208 × 184 grid points) grid increments, respectively."

Comment 3:
Why was event #2 selected for the calibration among the other events? Please add more details on the structure/scale of these events – were they all microscale, mesoscale or synoptic situations? This has severe implications on the robustness of the conclusions which may be governed by the microphysics options rather than the WRF-Hydro coupling. The authors select the WSM6 microphysics scheme without providing any justification. Are their previous sensitivity studies done for Greece or the surrounding region to support this selection and its relevance to the simulated storm scale(s)?

**Concerning the events:**
The selection of events #2 and #5 is primary related to the capability of the model to reproduce the observed rainfall in the study catchments, as an accurate representation of the atmospheric forcing is important for the simulation of the stream discharges and, consequently, for the calibration process.
The description of the synoptic conditions related to the examined events has been updated in lines 127-154.
**Lines 127-154:**
"…Six flood events have been considered for the analysis. Table 1 includes the simulation periods of each event, which were selected after spin-up sensitivity experiments (section 2.2.1), and their observed total rainfall and maximum discharge as they have been recorded at the meteorological and hydrometric stations. All examined episodes were associated with synoptic atmospheric circulation, driven by low-pressure systems, which, in most cases, were combined with 500-hPa troughs and cut-off lows. In particular, surface low-pressure systems, found west of Greece, affected the country in combination with upper-level cut-off lows on 6 February 2012 (event #3) and 29 December 2012 (event #4). In the course of events 1 and #6, the atmospheric circulation was characterized by troughs in the middle troposphere over Greece, associated with surface cyclones located west of North Italy (event #6) and in the Ionian Sea (event #1). The systems induced considerable precipitation in Greece during the above episodes resulting to noticeable impacts over the examined basins (Giannaros et al., 2020). The higher impacts in Sarantapotamos catchment were reported in Vilia at the night between 21 and 22 February 2013 (event #5), when 24-h precipitation and maximum discharge reached up to 77 mm and 19.2 $m^3$/s, respectively. During this episode, a very deep surface low crossed the Mediterranean Sea towards Greece. The system was associated with an upper-level trough having a negatively titled axis (Giannaros et al., 2020). Between 02 and 05 February 2011 (event #2), exceptional atmospheric conditions affected Greece (Giannaros et al., 2020). Significant impacts were evident in Rafina catchment where the total 48-h rainfall surpassed 123 mm in N. Makri and the maximum discharge exceeded 24 $m^3$/s in Rafina. As highlighted above, the events #2 and #5 affected the examined areas more severely and were the

most devastating for the whole area of Attica, where floods, deaths, destruction and great economic losses were induced. More details on the hydrometeorological and socio-economic characteristics of events #2 and #5 can be found in Giannaros et al. (2020)."

**Concerning the model configuration**, several preliminary tests have been performed in the framework of setting up the model for operational forecasting in Greece. The manuscript was modified to clarify the above and justify the selection of physics parameterizations (lines 175-181)
**Lines 175-181:**
"…The selection of the physics schemes was based on sensitivity tests conducted for the exploration of the best-performing schemes in terms of precipitation forecasting in the framework of setting up the model for operational forecasting in Greece. For the cloud microphysics processes, the WRF Single-Moment 6-Class Microphysics scheme (WSM6; Hong and Lim, 2006) was used, which has been also implemented in other studies over Greece (e.g. Emmanouil et al., 2021; Politi et al., 2018; Giannaros et al., 2016; Pytharoulis et al., 2016)."

**Minor comments/corrections**
Line 145: please justify the selection of the NOAH LSM instead of the NOAH-MP LSM (also comment on the selection of the MYJ PBL scheme vs. other schemes).

The above suggestions have been applied to the manuscript at lines 185-190.
**Lines 185-190:**
"…Noah-MP introduces multiple options and tunable parameters to simulate the land surface processes. However, the default values of these options and parameters are not suitable for every study area (e.g. Giannaros et al., 2019). In contrast, the Noah LSM has been tested and applied successfully in several studies focusing in Greece (e.g. Varlas et al., 2019; Papaioannou et al., 2019; Giannaros et al., 2020). In addition, MYJ parameterization scheme has been successfully implemented in other studies over Greece (e.g. Emmanouil et al., 2021; Politi et al., 2018)."

Line 8 (abstract): This study presents an integrated modeling approach for simulating flood events.
Line 12: Remove "on the improvement of"
Line 14: carried out with "the" WRF-Hydro model. There should also be mention of the comparison with WRF-only (standalone/uncoupled) runs.
Line 26: …especially "in its capital, Athens," flooding events…
Line 51: revise to "WRF-Hydro is a recently developed coupled hydrometeorological system that has been used for numerous research applications
Line 61: remove "the" before 36%
Line 75: add "the" before Cithaeron
Line 86: revise to "In the current study, we focus on two…"
Line 89: replace "intense" with "increasing" before urbanization
Line 100-103: capitalize "H" in "WRF-hydro" and correct the sentence structure.
Line 106: "Namely" is used incorrectly here
Line 113: add of: "...the whole of Greece…"
Line 137: add for "…of the area for better simulation…"
Line 218: Use either the long dash (–) or short dash (-) concisely for the term Nash-Sutcliffe

All the above issues have been addressed in the manuscript, as the reviewer suggested

Line 140: please justify the selection of WSM6 MP scheme for the study domain. Are their sensitivity studies done for Greece or the surrounding region to support this selection?

Please refer to the main comment concerning the microphysics scheme.

Figures:
Merge figures 5 and 6 using subplots and add error metrics on each subplot
Merge figures 9, 10 and 11 using subplots and add error metrics on each subplot

We have modified the figures 5,6, 9,10 and 11 according to the suggestions

---

## Author Response (AR2)

I thank the authors for carefully considering my comments. I acknowledge that the quality of the paper has improved to a certain extent. However, I still have some doubts, which I express starting from my previous comments.

Introduction: in my opinion, it's not only a matter of listing a series of studies using WRF-Hydro for reproducing/analysing flood events, rather the authors should better contextualize their research in the framework of those applications, particularly in the Mediterranean area. I ask for further effort in providing reasoned literature background.

The introduction was modified in order to highlight the findings of all recent hydro-meteorological studies that used WRF-Hydro model focusing in the Mediterranean area (lines 61-70).

**Lines 61-70:**
"… Especially, in the Mediterranean area, Senatore et al. (2020) implemented WRF-Hydro in a catchment of Italy in order to highlight the impact of SST in operational forecasts. Avolio et al. (2019) showed that WRF-Hydro was capable to simulate the hydro-meteorological impact of a highly intensity rainfall event in Italy. Senatore et al. (2015) studied the impact of fully-coupled WRF-Hydro model in the forecasting precipitation and showed that the coup[led model provides improved simulation precipitation compared to those provided by WRF-only simulations. Furnari et al. (2020) showed that the implementation of WRF-Hydro has the potential to improve by up to 200% the precipitation forecasts over a small catchments area in Italy. Camera et al. (2020) has implemented WRF-Hydro in small catchment areas in Cyprus and showed how the accuracy of the input precipitation can strongly affect the hydrological simulation. Furthermore, Varlas et al. (2019) and Papaioannou et al. (2019) have showed that WRF-Hydro demonstrate adequate skill in reproducing two past flood events in Greece."

Calibration methods: LL 259-60 and 297-98: I still think that the authors should explore what happens if they relax the calibration boundaries.

The selected boundaries for the calibration in this study are the boundaries that have been used in several previous studies available in the literature (e.g. Kerandi et al., 2018; Naabil et al., 2017; Givati et al., 2016; Yucel et al., 2015). The authors acknowledged that a possible relaxation of the boundaries might result to slightly better calibration results but as the calibration process in this study is performed manually, further exploration of the calibration boundaries would require almost 40 additional simulations for each event provided in this paper. Unfortunately the current computational availability does not allow us to perform these experiments. But certainly this is an issue that deserves further study in the future. Your important comment was addressed as follows in the text (lines 311-314):

**Lines 311-314:**
"For the needs of this study the range of 0.5 to 1.5 was used as this same range was proposed in the literature by previous studies (Kerandi et al., 2018; Naabil et al., 2017; Givati et al., 2016;

Yucel et al., 2015). However lower values than the lower boundary of 0.5 might provide improved results and this issue deserves further investigation in the future"

Results:
Though the soil moisture state immediately before the event is important, I think that concerning latent heat flux the analysis should be made in terms of accumulated values (at least, let's say, starting one day before). However, most importantly, the selected events are characterized by a strong shoreward flux of humid air, which most probably almost hide the effect of the land surface as a moisture source. What I mean is that moisture contribution from the land surface during the events is a second-order process, compared to sea surface contribution. I think that the authors should at least discuss this point.

The analysis concerning the heat flux was made in terms of accumulated values, as the reviewer suggested. The temporal period of accumulation was specified 6 hours before the beginning of the event, as for some events the spin-up period was less than 1 day. The manuscript at lines 362-365 and the results at the table 7 were modified accordingly.

**Lines 362-365:**
"…Table 7 shows the basin average soil moisture (at the 1$^{st}$ level) and 6 hour accumulated latent heat flux simulated by the WRF-Hydro and WRF-only models, at the time before the beginning of the examined storms events. As can be seen the soil moisture differences between the models range from 0.005 to 0.027 m$^3$ m$^{-3}$ and accumulated latent heat flux differences span from 4.1 to 41.8 W/m$^2$…."

Table 7

| | Basin | | Soil moisture (m$^3$ m$^{-3}$) | Accumulated latent heat (W/m$^2$) |
|---|---|---|---|---|
| Event #1 /E1 | Rafina | WRF-Hydro | 0.2915 | 1.4 |
| | | WRF | 0.3034 | -2.7 |
| Event #2 /E2 | Rafina | WRF-Hydro | 0.2760 | 40.1 |
| | | WRF | 0.2660 | 39.3 |
| Event #3 /E3 | Rafina | WRF-Hydro | 0.3427 | 388.1 |

| Event | Location | Model | | |
|---|---|---|---|---|
| | | WRF | 0.3159 | 346.3 |
| Event #4 /E4R | Rafina | WRF-Hydro | 0.2126 | -29.3 |
| | | WRF | 0.2121 | -29.1 |
| Event #4/E4S | Sarantapotamos | WRF-Hydro | 0.2248 | 235.2 |
| | | WRF | 0.2316 | 225.7 |
| Event #5 /E5 | Sarantapotamos | WRF-Hydro | 0.2834 | -9.4 |
| | | WRF | 0.2823 | -10.7 |
| Event #6 /E6 | Sarantapotamos | WRF-Hydro | 0.2792 | 20.3 |
| | | WRF | 0.2666 | 10.5 |

Concerning the land-sea interaction the manuscript was modified in order to highlight further this (lines 373-375).

**Lines 373-375:**
"…Furthermore, the soil moisture is strongly dependent to the sea-atmosphere interactions (Avolio et al., 2019; Senatore et al., 2015) and the synoptic scale circulation…"

Concerning the use of ERA5 reanalyses, in principle, they do not resemble either the reliability of 2011-2014 operational forecasts or current operational forecasts. Of course, I don't ask for considering more recent events with higher resolution GFS forecast, however, in my opinion, the authors have two options: 1) using 0.5° resolution GFS forecasts analyzing the "operational forecasting purposes" at that time; 2) using ERA5 reanalyses and re-modulating the discussion acknowledging that the results achieved are only partially informative concerning operational forecasts. A discussion about the difference in accuracy/reliability of reanalyses/real-time operational boundary conditions could provide a more complete picture of the problem.

The presented analysis was performed using ERA5 data as:

1) The available GFS data for this historical period were not adequate for forcing the WRF simulations in terms of spatial resolution
2) We have not access to operational ECMWF IFS forecasts
3) The ERA5 data have been utilized in several hydrological studies, indicating that they are adequate for hydrological modeling applications
Thus, we used the best available data sources in order to make an attempt to build a flood forecasting system based on the WRF-Hydro model.
Currently, the WRF-Hydro forecasting model is implemented in a pre-operational mode forced by GFS operational data (which is now 0.25°). A follow-up study could focus on evaluating the performance of the model during this pre-operational application.
The manuscript has been modified accordingly in order to clearly specify the above points in the methodology (lines 177-178) and in the results (lines 429-434).

**Lines 177-178:**
"…Furthermore, the ERA5 reanalysis dataset has been proved reliable for hydrological modeling applications (Alves et al., 2020; Crossett et al., 2020; Martens et al, 2020; Tarek et al., 2019)…"

**Lines 429-434:**
"…A follow-up study could focus on evaluating the performance of the model initialized by GFS data during a pre-operational application covering a whole hydrological year. Such a study could enhance our knowledge about the added value of the WRF-Hydro coupled mode and shed light on the performance of the model using GFS operational data. The utmost goal is to provide citizens and stakeholders with reliable information and warnings in order to enhance flood risk awareness and protect lives and properties.…"

Conclusions: I suggest highlighting better why the fully-coupled WRF-Hydro operational forecasts should be preferred to the one-way coupled, notwithstanding the increased computational burden. Is the improvement worth it? The authors should make clear their reasoning (e.g., what kind of trade-off between accuracy and the computational burden they consider to get their conclusions).

It should be noted that the current study does not argue that the fully coupled WRF-Hydro should be preferred compared to the one-way coupled WRF-Hydro for operational application.
Our results indicate a beneficial effect of the coupled WRF-Hydro in simulating accurately the precipitation compared to the WRF-only. This is important for the rainfall-dominated catchments as the studied ones, when mostly precipitation drives the hydrological response.

**Anonymous Referee #2**

The authors have addressed all my comments and justified their work as a preliminary attempt to develop a flood forecasting system based on the WRF-Hydro model. I only have a few minor comments for them to address before the manuscript can be accepted.

Line 46: also cite the original work which diagnoses the soil moisture-precipitation feedback by: Eltahir, E. A. (1998). A soil moisture–rainfall feedback mechanism: 1. Theory and observations. Water resources research, 34(4), 765-776.

Line 49: also cite more recent work on simulating extreme events using WRF-Hydro by:
-Wehbe, Y., Temimi, M., Weston, M., Chaouch, N., Branch, O., Schwitalla, T., Wulfmeyer, V., Zhan, X., Liu, J. and Mandous, A.A. (2019). Analysis of an extreme weather event in a hyper-arid region using WRF-Hydro coupling, station, and satellite data. Natural Hazards and Earth System Sciences, 19(6), 1129-1149.

- Pal, S., Dominguez, F., Dillon, M. E., Alvarez, J., Garcia, C. M., Nesbitt, S. W., & Gochis, D. (2020). Hydrometeorological Observations and Modeling of an Extreme Rainfall Event using WRF and WRF-Hydro during the RELAMPAGO Field Campaign in Argentina. Journal of Hydrometeorology.

Line 53: After mentioning WRF model for the first time, cite:
Skamarock, W. C., Klemp, J. B., Dudhia, J., Gill, D. O., Barker, D. M., Wang, W., & Powers, J. G. (2005). A description of the advanced research WRF version 2. National Center For Atmospheric Research Boulder Co Mesoscale and Microscale Meteorology Div.

We would like to thank the reviewer for the suggestions. All the above citations have been properly implemented in the manuscript (lines 44-50 and 52-54).

**Lines 44-50:**

"…The terrestrial hydrological processes affect soil moisture, a variable that is crucial for the computation of the sensible and latent heat fluxes, which in turn affect the atmospheric response (Seneviratne et al., 2010; Maxwell et al., 2007, Etalhir, 1998). Several studies have shown that an improvement, although not always significant, on the forecasting of the spatiotemporal distribution of extreme synoptic and convective precipitation is provided through the use of coupled hydrometeorological models (e.g., Pal et al., 2020; Wehbe et al., 2019; Senatore et al., 2015; Shrestha et al., 2014; Maxwell et al., 2007). .."

**Lines 52-54:**

"…WRF-Hydro, an enhanced version of the Weather Research and Forecasting (WRF; Skamarock et al., 2005) model, is one of the various modeling systems that provides a two-way coupling between the hydrological and land-atmosphere processes…"

Figure 1 caption: part of the text is italic.
The caption was corrected

Figure 9: please improve the quality of the figure and consider using more contrasting colors for WRF vs. WRF-Hydro.

Figure 9 was updated accordingly